# Multifaceted Roles of IL-26 in Physiological and Pathological Conditions

**DOI:** 10.3390/ijms27010325

**Published:** 2025-12-28

**Authors:** Boryana Georgieva, Danijela Karanović, Ivona Veličković, Danail Minchev

**Affiliations:** 1Department of Human Anatomy and Physiology, Faculty of Biology, Paisii Hilendarski University of Plovdiv, 24 Tsar Assen St, 4000 Plovdiv, Bulgaria; bgeorgieva@uni-plovdiv.bg; 2Institute for Medical Research, National Institute of Republic of Serbia, University of Belgrade, Dr Subotića 4, 11000 Belgrade, Serbia; 3Institute of Botany and Botanical Garden “Jevremovac”, University of Belgrade-Faculty of Biology, 11000 Belgrade, Serbia

**Keywords:** IL-26, inflammation, antimicrobial activity, cancer

## Abstract

Cytokines are a diverse group of signaling proteins that regulate immune responses by mediating cell communication. Among them, interleukins (ILs) play essential roles in immune regulation, influencing diverse cell processes through tightly controlled signaling networks. Dysregulation of interleukin signaling could lead to chronic inflammation, contributing to the development of autoimmune and inflammatory diseases as well as cancer. IL-26, a cytokine of the IL-10 family, has emerged as a unique modulator of immune function. Although structurally related to IL-10 and sharing one of its receptor subunits, IL-26 exerts distinct biological effects, particularly in promoting inflammatory responses and interacting with extracellular DNA to activate immune pathways. Increasing evidence implicates IL-26 in the development of several chronic conditions, such as psoriasis, rheumatoid arthritis, inflammatory bowel disease, asthma, and various types of cancer. This review summarizes current knowledge on IL-26’s biology, including its structural and receptor characteristics, immunomodulatory functions, and roles in inflammation and disease. Understanding IL-26’s dual functions in normal and inflammatory states may provide insights into novel therapeutic strategies targeting IL-26-mediated pathways in pathological conditions.

## 1. Introduction

Cytokines are a diverse group of protein factors, consisting of chemokines, interferons, interleukins, lymphokines, and tumor necrosis factors, that mediate and regulate cell interactions, especially in the context of inflammatory and immune responses. A large group of cytokines, called interleukins (ILs), play essential roles in regulating immune responses. Cellular responses to ILs encompass a multitude of up- and down-regulatory mechanisms that also involve genes for inhibitors of the cytokine receptors. ILs may influence the synthesis and functioning of other ILs. For instance, IL-1 induces lymphocyte activation and the secretion of IL-2 [1]. Cytokines exert either autocrine or paracrine effects on the same and nearby cells or influence distant cells through the bloodstream. Because the binding between interleukin molecules and their receptor is strong, a relatively low concentration of cytokine molecules is required to engage their receptors and trigger downstream biological responses [2,3]. ILs play important roles in controlling immune cell activity and regulating their proliferation, maturation, migration, and adhesion. They possess both pro- and anti-inflammatory properties [4,5]. Their primary role is to modulate growth, differentiation, and activation in the context of inflammatory and immune responses. Inflammation is a complex physiological process crucial in host defense and tissue repair. However, dysregulated inflammation can lead to chronic inflammation. This persistent inflammatory state plays a central role in various conditions, including many autoimmune disorders and cancer.

IL-26 belongs to the IL-20 subfamily of the IL-10 family together with IL-19, IL-20, IL-22, and IL-24 (Table 1). Although it shares structural similarities with IL-10 and interacts with one of the subunits of the IL-10 receptor [6,7], IL-26 possesses distinct functional properties that differentiate it from IL-10.

IL-26 is emerging as a key player in regulating immune cell activity and inflammatory signaling pathways. Despite its relatively recent discovery, IL-26 has been documented to play a role in various chronic conditions. These include cancer, heart disease, rheumatoid arthritis (RA), asthma, chronic obstructive pulmonary disease (COPD), Crohn’s disease (CD), nonspecific ulcerative colitis (UC), atopic dermatitis (AD), and even Alzheimer’s disease [14,15,16].

The purpose of this review is to summarize recent findings on the potential pathological roles of IL-26 in inflammatory disorders and cancer. It aims to highlight the essential role of IL-26 in inflammation, both in norm and in pathology.

## 2. Genomic Organization and IL-26 Protein Properties

IL-26 was first identified by Knappe et al. [17], who used subtractive hybridization to characterize a novel cDNA, termed ak155. It was specifically expressed in human T cells transformed by subgroup C strains of herpesvirus saimiri (HVS). The new transcript showed weak nucleotide homology to the cellular interleukin-10 (IL-10) gene. Northern blotting and RT-PCR analysis revealed strong expression of ak155 in HVS-transformed human T-cell lines and low-level expression in unstimulated peripheral blood cells from healthy humans. Initial functional analyses showed that recombinant AK155 forms dimers, both when expressed in *Escherichia coli* and when expressed in eukaryotic cells. The gene was mapped to chromosome 12q15 at a locus later defined as IL-26.

The genomic cluster (IFNγ-IL-26-IL-22) is conserved and transcribed in the same orientation across several vertebrate species, including fish, birds, and amphibians, but is notably absent in mice and rats, despite the presence of IL-26 receptor genes [6,18,19,20].

IL-26 comprises 171 amino acids and has a molecular weight of ~19 kDa in monomeric form. In vivo, IL-26 typically forms homodimers (~36 kDa) and can assemble into higher-order multimers [21]. Structural modeling has shown that IL-26 multimers may adopt an elongated ‘beads-on-a-string’ conformation, a configuration that is unusual within the IL-10 family [21]. IL-10 family cytokines share a conserved structural framework, typically composed of six or seven α-helices connected by loops [12,22,23,24,25]. Unlike other IL-10 family cytokines, IL-26 is a highly cationic and amphipathic molecule, containing 30 positively charged amino acids with an isoelectric point of ~10.4–10.7 [17,26,27] and a net charge of +19.5 at physiological pH. These properties contribute to its ability to bind nucleic acids and glycosaminoglycans [28].

Three-dimensional modeling predicts that IL-26 adopts a compact bundle of six α-helices stabilized by two intramolecular disulfide bonds (Figure 1). Four conserved cysteine residues in the IL-molecule appear to be important for dimer formation [6]. Helices E and F are amphipathic, with positively charged hydrophilic faces that are enriched in lysines and arginines, and hydrophobic faces that facilitate membrane interaction [21,25]. These features provide IL-26 with properties similar to cationic cell-penetrating peptides (CPPs), which are known to shuttle extracellular DNA into cells [29].

No post-translational modifications of IL-26 have been conclusively characterized to date. However, differences in activity have been observed between naturally produced and recombinant forms of IL-26. For instance, IL-26 secreted by Th17 cells shows greater antibacterial activity than recombinant IL-26 produced in prokaryotic systems, suggesting that post-translational modifications or synergistic interactions with other immune molecules may enhance its function [21,30].

## 3. In Silico Methods in Studies of IL-26

Key applications of in silico methods for characterizing IL-26 include predicting structure and physicochemical properties, reconstructing interaction pathways, integrating omics data, identifying expression quantitative trait loci, and exploring gut homeostasis. A sequence alignment using the ClustalW software revealed a higher similarity between IL-26 and IL-10 (26%) compared to IL-19 (20%) and IL-22 (21%). Thus, a structural model of IL-26 has been developed by in silico modeling with the existing IL-10 crystal 3D structure used as a starting point [25]. The proposed 3D layout of the IL-26 molecule was obtained using Modeler software 9v2 and subsequently verified by Procheck. Further in silico analyses were performed to determine physicochemical properties of the IL-26 molecule, like surface charge densities (Adaptive Poisson-Boltzmann Solver), putative DNA-binding sites (MetaDBsite), and amphipathic helices (AmphipaSeek) [25]. A comprehensive map of IL-26 signaling has been constructed involving 7 activation or inhibition interactions, 16 catalytic steps, 33 gene expression control points, 25 protein expression types, 2 transport mechanisms, and 3 molecular associations [31]. A genome-wide genotyping has identified cis-expression quantitative trait loci associated with IBD-risk genes and immunity. Among them, an inflammation-dependent polymorphism, rs12582553, is located in close proximity to the IL-26 gene [32]. A recent study integrated genomics and radiomics data using machine learning algorithms. This led to the development of a predictive model for colorectal cancer metastasis, in which the expression of IL-26 at both the mRNA and protein levels serves as an essential marker [33].

## 4. Membrane Receptors

The canonical IL-26 receptor is composed of IL-20 receptor alpha (IL-20R1) and IL-10 receptor beta (IL-10R2) subunits [6]. IL-20R1 provides the ligand-binding domain, while IL-10R2 is essential for receptor complex assembly, which activates multiple downstream signaling pathways. IL-10R2 is broadly expressed across hematopoietic and non-hematopoietic cells and shared among multiple cytokines, including IL-10, IL-22, and interferons [22]. IL-20R1 expression is limited in epithelial cells, keratinocytes, and some myeloid cells such as monocyte-derived dendritic cells [7,34,35]. It is also expressed in non-hematopoietic cells, like those in lungs, placenta, and reproductive tissues [6,10].

## 5. Receptor-Independent Mechanisms

IL-26 activates immune cells that do not express IL-20R1, such as monocytes, natural killer (NK) cells, and B cells [14,36,37]. IL-26 also induces functional responses in structural cells like endothelial and triple-negative breast cancer cells (TNBC), which lack detectable IL-20R1. These findings collectively indicate a non-classical entry mechanism and further support the involvement of receptor-independent or alternative receptor-mediated signaling [38,39].

The mechanisms by which IL-26 affects cells that lack conventional membrane receptors are still a subject of research. Observations that monomeric and dimeric IL-26 differ in their capacity to activate IL-26R-expressing epithelial cells, compared with IL-20R1-deficient immune cells [14], indicate the existence of distinct signaling pathways [14]. One suggested mechanism is that IL-26 can enter cells independently of its classical receptor through interaction with negatively charged surface molecules such as glycosaminoglycans on the plasma membrane (e.g., heparin, heparan sulfate) [21,25,40,41]. An alternative mechanism suggests that helix F contains an in-plane membrane (IPM) motif anchor [25], a structural domain known to mediate protein binding to cell membranes and facilitate internalization [42]. Additionally, IL-26 exhibits characteristics of CPPs, enabling nonspecific membrane translocation [43].

IL-26 has a unique ability to bind extracellular DNA released by dying cells, a property particularly relevant in chronic inflammatory and autoimmune diseases. IL-26 binds various forms of DNA, including genomic DNA, mitochondrial DNA, and neutrophil extracellular traps (NETs). While NETs alone do not elicit strong inflammatory responses, IL-26 confers immunogenicity by facilitating the entry of DNA into immune cells [25]. Immunofluorescence microscopy and solid-phase binding assays confirmed interaction of IL-26 with chromatin structures within NETs [25].

In plasmacytoid dendritic cells (pDCs), which lack expression of IL-20R1, IL-26-DNA complexes are internalized via cell-surface heparan sulfate proteoglycans (HSPGs), leading to toll-like receptor 9 (TLR9)-mediated production of IFN-α. These effects mirror other known DNA-carrier molecules such as LL37 and HMGB1, which enhance DNA uptake and immune recognition in diseases like systemic lupus erythematosus (SLE) and psoriasis [21].

To elucidate the IL-26-mediated DNA uptake in pDCs, Meller et al. [21] initially applied Alexa Fluor 488-labeled DNA to trace IL-26–DNA complexes into pDCs. The authors observed that IL-26–DNA complexes, but not naked DNA, interact with pDCs. Furthermore, this interaction was strongly dependent on the IL-26 dose. In addition, the fluorescent signal accumulated predominantly in intracellular vesicular structures within the target pDCs. Confocal microscopy revealed that these structures were enriched for the early endosomal marker CD71, suggesting they most probably resulted from endocytosis. The internalization of IL-26–DNA markedly decreased at low temperatures (4 °C) or after the application of the inhibitor cytochalasin D, further confirming that IL-26–DNA internalization occurred via endocytosis. The association between IL-26 and DNA was disrupted by anionic heparin in a concentration-dependent manner, suggesting that the IL-26–DNA interaction relies on the positively charged amino acids of IL-26. These findings prompted further investigation into the mechanism of the IL-26–DNA uptake in pDCs. For instance, heparan sulfate proteoglycans (HSPGs) serve as receptors for various macromolecules that are transferred across membranes. In the case of IL-26–DNA endocytosis, Meller et al. [21] observed that HSPGs are essential for the internalization of the complexes, as confirmed by the administration of trypsin and heparinase III to pDCs. Following endosomal entry, IL-26–DNA elicited specific effects in pDCs, including IFN-α secretion, through the activation of TLR9. Unsurprisingly, this IFN-α production was inhibited by chloroquine, suggesting the essential role of TLRs in pDC activation. Treatment of pDCs with cGAMP could rescue IFN-α production in the case of chloroquine inhibition, further confirming the role of STING in the IL-26–DNA-triggered IFN-α secretion. Moreover, the effect of IL-26–DNA complexes was shown to be independent of IL-10R2 or STAT3 and, at least in part, dependent on NF-κB activation [21].

In addition, IL-26 exhibits several properties of CPPs. These properties include two helices of amphipathic nature (E and F), the presence of cationic amino acid residues, and an IPM anchor, all of which have the potential to facilitate its function as a DNA-internalizing agent [25]. In silico analyses predict that IL-26 contains DNA-binding surfaces located on the E and B helices, as well as in the N-terminal region of the C helix. Among the IL-26 α-helices (A–F), helices E and F demonstrate amphipathic properties, as predicted by in silico models. They possess both hydrophobic and hydrophilic surfaces, the latter of which contains several positively charged amino acids. DNA-binding and amphipathicity are key properties of CPPs engineered for DNA transfer, such as the synthetic KALA peptide [29]. Common features of the F helix of IL-26 and the KALA peptide include an α-helical structure with a hydrophilic surface rich in lysine residues and a hydrophobic surface possessing leucine residues. Additionally, the F helix contains a predicted IPM motif, capable of anchoring proteins to membranes. Detailed functional analyses have suggested that IL-26–DNA complexes stimulate the activation of myeloid cells in both inflammasome-dependent and independent manners [25,44], most probably via TLR9/STING signaling. As a result, extracellular DNA facilitates the engagement of both endosomal and cytosolic DNA sensors, notably the cGAS–STING axis and the AIM2 inflammasome, and triggers robust proinflammatory responses [25]. These complexes act as potent immunostimulatory agents driving the activation of innate immune cells such as monocytes and pDCs [21]. In myeloid cells, this activation leads to the production of type I interferons, particularly IFN-β, as well as proinflammatory cytokines such as IL-6, IL-1β [14], and IL-8 [45]. Importantly, these immune responses occur independently of classical IL-26 receptor signaling.

Whether human monocytes express TLR9 remains a subject of debate. In accordance with the absence of TLR9 in human monocytes, it has been demonstrated that bafilomycin A, an inhibitor of TLR signaling, does not suppress monocyte and neutrophil activation by IL-26–DNA complexes [46,47]. By contrast, a STING knockdown attenuated the response to IL-26–DNA and reduced both cGAMP and DNA sensitivity in THP1 ISG cells [25]. In addition, siRNA-mediated silencing of STING markedly reduced IL-6 secretion by monocytes in culture following IL-26–DNA stimulation. Furthermore, it has been demonstrated that IL-26 and IL-26–DNA complexes induce phosphorylation of IFN regulatory factor 3, which participates in signaling transduction downstream of STING [25]. Overall, the proposed mechanism in pDCs involves interaction with surface proteoglycans and TLR9 activation. In contrast, the internalization of the complexes may elicit monocyte activation independently of TLR9 inhibition. Regardless of these discrepancies, the available experimental data establish the essential role of STING signaling in maintaining the capacity of IL-26–DNA complexes to elicit a response in both monocytes and pDCs.

The pathophysiological relevance of IL-26-DNA complexes is supported by elevated levels found in patients with autoimmune diseases, including psoriasis, SLE, and anti-neutrophil cytoplasmic antibody-associated vasculitis (AAV) [25,38,48]. These conditions share features of heightened cell death, impaired DNA clearance, and chronic type I IFN responses, all of which create an environment conducive to IL-26-mediated immune activation. Patients with active AAV exhibit elevated levels of circulating IL-26 and IL-26-DNA complexes, particularly during acute disease flares [25]. These findings suggest a possible mechanism in which IL-26-DNA complexes present in inflammatory lesions may initiate a self-amplifying cycle in which excessive cell death promotes sustained inflammation and vice versa. Notably, IL-26-DNA complexes promote not only cytokine production but also protease expression in monocytes and neutrophils, further contributing to tissue damage and inflammation [49].

Collectively, IL-26 acts as a DNA-binding immunostimulatory factor that bridges tissue damage with innate immune activation and sustained inflammatory responses. Through its ability to transform extracellular self-DNA into a proinflammatory signal, IL-26 emerges as a critical player in the pathogenesis of inflammatory diseases.

## 6. Cellular Source of IL-26

IL-26 mRNA is expressed predominantly in activated T cells, especially Th17 cells. Human T helper 17 (Th17) cells are a specialized subset of CD4^+^ T cells that mediate inflammatory responses by producing cytokines such as IL-17A, IL-17F, IL-21, IL-22, and IL-26 [50,51,52]. IL-26 is now recognized as a pleiotropic molecule produced by a broad range of immune and structural cells. Aside from Th17 cells, these include CD8^+^ T cells, NK cells, macrophages, neutrophils, fibroblasts, and epithelial and smooth muscle cells [16,26,50,53,54,55,56,57,58].

In the lungs, bronchial epithelial cells, alveolar macrophages, CD4^+^ and CD8^+^ T cells, as well as lung fibroblasts constitutively express IL-26 [56,59,60,61]. Bronchial epithelial cells produce IL-26 in response to stimulation with viral mimetics (poly I:C, imiquimod, ssRNA) and proinflammatory cytokines (IL-1β, IL-17A, TNF-α), with enhanced effects in the presence of IL-22 [56]. Similarly, lung fibroblasts release IL-26 in response to endotoxin via MAPK, NF-κB, and TRIF-dependent pathways, which are inhibited by asthma and COPD therapies such as hydrocortisone, salbutamol, and tiotropium [61]. Alveolar macrophages, critical for airway immunity, both produce and respond to IL-26. They express both receptors, IL-10R2 and IL-20R1, and downstream effectors, STAT1 and STAT3, suggesting the presence of autocrine or paracrine feedback loops [10,57]. Tobacco smoke and bacterial endotoxin further enhance IL-26 release from these cells [57].

NK cells, particularly the IL-22-producing NKp44^+^ ILC3 subset, are also capable of producing IL-26 [26,62]. This extends to immature CD117^+^CD161^+^ NK cells in mucosal-associated lymphoid tissues [63]. Zhang and colleagues confirmed IL-26 expression in NK and NK T cells from tuberculosis patients [64].

Neutrophils released IL-26 in response to stimuli such as endotoxin or *Klebsiella pneumoniae* [65]. These cells also express STAT1 and STAT3, indicating responsiveness to IL-26 [66].

Expression is minimal in B cells, although herpesvirus-transformed B cells can produce low levels [17]. Conflicting data exist about the expression of IL-26 in monocytes [34,67].

The memory Th1 cells are also a source of IL-26 [26,34,55]. Regulatory T cells and Th2 cells, by contrast, express little or no IL-26 [34]. IL-26 production correlates with IL-17A expression and can be induced by IL-1β, IL-23, and RORγt overexpression [50,51,68]. IL-26^+^ T cells are enriched in inflamed tissues in inflammatory bowel disease (IBD), RA, psoriasis, and chronic lung conditions [14,50,55,69]. IL-26 expression also occurs in mucosal-associated invariant T cells (MAIT) [58,70].

Fibroblast-like synoviocytes (FLS) in RA patients produce high levels of IL-26, enhanced by IL-1β and IL-17A [14]. Unlike healthy FLS, in a state of RA, FLS constitutively produce IL-26, and their stimulation increases cytokine output. CD3^+^ T cells, particularly RORγt^+^ subsets, also contribute to IL-26 production in the inflamed synovium [55].

Across other non-immune cells that express IL-26 are smooth muscle cells and colonic myofibroblasts [16,53,56]. Renal arterial smooth muscle cells have also been shown to express IL-26 in ANCA-associated vasculitis [25]. However, its expression is absent in keratinocytes, endothelial cells, and skin fibroblasts under basal conditions [34].

Table 2 and Figure 2 provide a concise summary of the various cell types expressing IL-26.

Several studies have reported conflicting results regarding IL-26 expression in monocytes. For instance, two studies by Wolk et al. have suggested IL-26 expression is absent in monocytes [34,73]. However, according to other sources, IL-26 expression in monocytes is constitutive, although it is lower than in T cells [67]. When considering activated monocytes specifically, IL-26 exhibits reduced expression in monocytes infected with *Mycobacterium tuberculosis* at the RNA level [67]. Conversely, in monocytes exposed to lipopolysaccharides (LPSs) and IFN-γ, combined with an anti-IL-10 antibody, the production of IL-26 increases [74]. Furthermore, alveolar mature macrophages isolated from the lungs of healthy individuals secrete IL-26 as a response to endotoxin exposure [59]. Existing discrepancies across various studies may be potentially explained by different protocols for sample processing, dissimilar strategies for inducing cell activation, and, especially, different detection methods. For instance, in both studies by Wolk et al. [34,73] and in the work of Guerra-Laso et al. [67], the authors apply magnetic cell separation to purify CD14^+^ monocytes from white blood cell populations, while Nagalakshmi et al. [74] utilize elutriation for the purpose of monocyte isolation. Meanwhile, Che et al. [59] preferred immunocytochemistry and immunocytofluorescence combined with fluorescence-activated cell sorting to isolate and characterize mature CD68^+^ macrophages from bronchoalveolar lavage samples. Discrepancies in the stimulation/differentiation procedures are also significant confounding factors: LPSs from *Escherichia coli* [34], culturing in the presence of M-CSF [73], infection with *Mycobacterium tuberculosis* [67], LPSs and IFN-γ paired with an anti-IL-10 antibody [74], and endotoxin stimulation [59]. However, the factor of primary importance potentially affecting the rigor of experiments is the method of detection. For example, several studies rely only on RT-qPCR as a technique for quantitative evaluation of IL-26 levels [34,73,74]. However, applying more than one quantitation method, which produces comparable results, strengthens the reliability of the data obtained. Accordingly, IL-26 has been characterized in blood monocytes in the context of *Mycobacterium tuberculosis* infection by microarrays, RT-qPCR, and ELISA, confirming the constitutive expression of IL-26 and its downregulation as a response to Mycobacterium infection [67]. Meanwhile, the study by Che et al. provides solid evidence for the expression of IL-26 in mature alveolar macrophages based on the marked colocalization of IL-26 and CD68 after immunocytochemistry, immunocytofluorescence, and fluorescence-activated cell sorting. These results have received further confirmation by ELISA and RT-qPCR [59].

## 7. Target Cells

IL-26 affects multiple cell types and exerts a plethora of biological functions (Figure 3). Given its proinflammatory potential, IL-26 expression is tightly regulated to prevent chronic pathology. Its expression is closely linked with Th1 and Th17 immune responses and shares regulatory elements with IFN-γ and IL-22, reflecting its integration into broader immune regulatory networks [17,75]. It acts as a bridge between innate and adaptive immunity, capable of modulating inflammatory responses in both immune and structural compartments.

Corvaisier and colleagues confirmed that IL-26 stimulates human monocytes to produce the proinflammatory molecules IL-1β, IL-6, and TNF-α in the context of RA. Moreover, increased synthesis of IL-1β by monocytes results in an elevated Th17 cell frequency [14]. Additionally, IL-26 may serve as an early marker of Th17 differentiation and sustain inflammation by promoting IL-17A and IL-23 production in memory T cells [14,76].

In non-hematopoietic cells, like keratinocytes and epithelial cells, IL-26 may act in an autocrine or paracrine manner, potentially amplifying inflammation through positive feedback on IL-26 receptor expression [59].
Figure 3Target cells of IL-26. Target cells include neutrophils (Mizuno et al., 2022) [49], M1 macrophages (Lin et al., 2020) [77], monocytes and memory CD4^+^ T cells (Corvaisier et al., 2012) [14], dendritic cells (Hawerkamp et al., 2020) [78], vascular endothelial cells (Hatano et al., 2019 [79]), keratinocytes and epithelial cells (Hör et al., 2004) [7], chondrocytes (Chen et al., 2021) [80], fibroblast-like synoviocytes and osteoclasts (Lee et al., 2019) [81], as well as subepithelial myofibroblasts (Fujii et al., 2017 [16]).
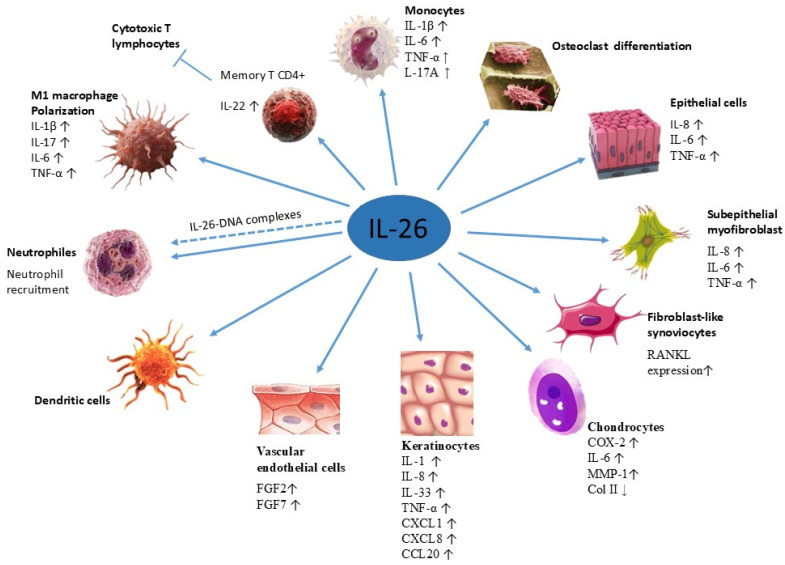


## 8. Functional Roles in Immunity

IL-26 acts as an upstream inflammatory mediator [53]. Upon ligand binding, IL-26 activates multiple signaling pathways, including JAK1 and TYK2 kinases, leading to phosphorylation of STAT1 and STAT3 [6,7,82]. Additionally, downstream cascades involving MAPKs (ERK, JNK), PI3K/Akt, NF-κB, and AP-1 are implicated in IL-26-mediated functions (Figure 4) [16,38,55].

In keratinocytes, IL-26 stimulation activates phosphorylation of STAT1, STAT3, JAK1, JAK2, and TYK2. This results in increased expression of CXCL8, IL1B, CCL20, IL33, and beta-defensin 4 (DEFB4). In murine models, IL-26 treatment enhances the expression of IL-4, -13, -33, -17A, and CCL20 in keratinocytes, thereby exacerbating the severity of AD [45].

Interestingly, exposure to recombinant human IL-26 in bronchoalveolar lavage samples has also been reported to exert regulatory effects, increasing STAT1 and STAT3 expression while concurrently suppressing MPO activity and reducing the expression of CXCL8, IL-1β, TNF, and CSF2 [59]. These changes are associated with modulated antibacterial host defense. Additionally, IL-26 induces the release of key neutrophil-activating cytokines, including IL-1β, TNF-α, and GM-CSF, from primary immune cells from endotoxin-primed human airways [59].

In hepatic stellate cells, IL-26 promotes cell activation and apoptosis by downregulating caspase-3 (CASP3), cleaved CASP3, and Bcl-2-associated X protein (BAX), while simultaneously upregulating IL-6, IL-10, TNF, MMP9, and actin alpha 2 (ACTA2). These effects are mediated via the TGF-β1/SMAD2 signaling pathway [64].

In monocyte/macrophage-like cells, IL-26 induces the phosphorylation of STAT1, c-JUN, and REL-A (NF-κB p65) in both the cytoplasm and nucleus. This signaling cascade leads to the upregulation of CD80, nitric oxide synthase 2 (NOS2), TNF, IL-6, and IL-10, indicating a role for IL-26 in both M1 and M2 macrophage polarization [77]. Lin and colleagues demonstrated that IL-26 directly promotes macrophage differentiation toward the proinflammatory M1 phenotype and can reprogram M2 macrophages previously polarized by M-CSF into M1-like cells. In their earlier work, they also reported that IL-26 inhibits macrophage differentiation into osteoclasts [83]. These findings indicate that IL-26 not only suppresses osteoclastogenesis but also actively influences macrophage polarization. In FLS derived from RA patients, IL-26 stimulation triggers a complex phosphorylation-dependent signaling network. This includes phosphorylation of IL-20R1, STAT1, STAT3, spleen-associated tyrosine kinase (SYK), MAPK1/3, MAPK8, MAPK14, IκBα (NFKBIA), and JUN. Downstream effects include increased expression of TNF superfamily member 11 (TNFSF11) and osteoclast-related markers, supporting the association of IL-26 with osteoclastogenesis and bone erosion [81]. Other studies also confirm that IL-26 influences bone homeostasis. As monocytes can differentiate into osteoclasts and osteoblasts in response to specific microenvironmental signals, IL-26 has been shown to inhibit the bone-resorbing function of differentiated osteoclasts [83] while enhancing bone mineralization by human osteoblasts [72], indicating a potential role in the regulation of bone remodeling. IL-26 promotes monocyte differentiation into proinflammatory M1-like macrophages, which produce cytokines such as IL-6 and TNF-α [77,83]. Wang and colleagues demonstrated that IL-26 can directly enhance IL-9 expression and regulate IL-17A production in macrophages in the context of RA. In agreement with the results of Lin and colleagues, they confirm that IL-26 exerts multiple functions in macrophages, including driving M1 macrophage polarization [84].

In pancreatic ductal adenocarcinoma cells, IL-26 engagement of its receptor complex induces phosphorylation of STAT3, MAPK1, and MAPK3, promoting tumor aggressiveness [85]. Similarly, in colorectal adenocarcinoma cells, IL-26 binding activates STAT1 and STAT3 phosphorylation at tyrosine residues, suggesting a proinflammatory and potentially tumorigenic role [6].

In line with the established involvement of IL-1β in human Th17 cell differentiation [86], IL-26-treated monocytes drive the polarization of non-committed memory CD4^+^ T cells toward the Th17 lineage [14]. IL-26 has also been reported to induce inflammatory mediator production in epithelial cells and to promote the differentiation of proinflammatory Th17 cells through monocyte-derived IL-17 and IL-23 [87]. Under these conditions, IL-26-induced Th17 cells also express IL-26, which may indicate the existence of a positive feedback loop that sustains Th17 polarization and inflammatory cytokine production, potentially contributing to the persistence of activated Th17 cells. Further studies have shown that IL-26-producing Th17 cells can comprise up to 30% of T lymphocytes infiltrating the lesions in psoriasis patients. Similar findings exist for RA and bronchial biopsies of patients with severe asthma [88].

In addition to promoting Th17 responses, IL-26 stimulates human monocytes to produce a broad array of chemokines that recruit both innate and adaptive immune cells [14]. Regarding adaptive immunity, Hummelshoj and colleagues reported that IL-26 inhibits immunoglobulin production by B cells, specifically reducing IgA and IgG secretion in response to anti-CD40 antibody stimulation, as well as suppressing IL-4-induced IgG4 production when combined with anti-CD40 antibody [36].

A marked increase in IL-26-expressing cells was observed in active CD lesions [55]. IL-26 can stimulate the production of matrix metalloproteinase 8 (MMP-8) and MMP-9 and inhibit MMP-1 in macrophages in IBD patients [89]. Findings by Fujii and colleagues [16] in colonic subepithelial myofibroblasts (SEMFs) demonstrated that the IL-26-activated cascade induces the transcription factors JUN and RELA (the NF-κB p65 subunit), resulting in the upregulation of IL-6 and C-X-C motif chemokine ligand 8 (CXCL8), both key mediators of mucosal inflammation.

In patients with asthma, overexpression of IL-26 is associated with elevated expression of IL-17A, RAR-related orphan receptor C (RORC), IL-1β, IL-6, and tumor TNF, driven by activation of the IL-20R1/IL-10R2 receptor complex [31,90]. These findings highlight a role for IL-26 in promoting Th17-associated inflammation in airway disease.

Itoh and colleagues [39] demonstrated that IL-26 interacts with ephrin receptor A3 (EPHA3), inducing phosphorylation of AKT1 and MAPK8 and suppressing epidermal growth factor receptor-tyrosine kinase inhibitor (EGFR-TKI)-induced endoplasmic reticulum stress in both murine and human TNBC cells. This pathway downregulates DNA damage-inducible transcript 3 (DDIT3) at the protein level, contributing to anti-apoptotic signaling and enhanced tumor progression. Although these findings presume the involvement of IL-26 in tumorigenesis, further investigation is needed to elucidate the broader implications of IL-26–EPHA3 signaling in cancer biology.

## 9. IL-26 as an Antimicrobial and Antiviral Cytokine

IL-26 displays direct antimicrobial effects and immunomodulatory capabilities in the defense against bacterial infections. Structurally and functionally, IL-26 shares features with antimicrobial peptides. IL-26 exerts direct antimicrobial activity by directly binding to the bacterial cell wall. Recombinant human IL-26 (rhIL-26) can kill both gram (−) bacteria (e.g., *Pseudomonas aeruginosa*, *Escherichia coli*, *Klebsiella pneumoniae*) and gram (+) bacteria (e.g., *Staphylococcus aureus*) by binding to lipopolysaccharide and lipoteichoic acid, essential components of bacterial cell walls, and inducing membrane pore formation and membrane blebbing, which lead to cytoplasmic leakage and bacterial death [21,65,91].

Beyond its direct antimicrobial role, IL-26 also boosts host defense by forming complexes with extracellular bacterial DNA. As noted above, these IL-26–DNA complexes induce the production of type I interferons by pDCs [21]. This process connects Th17-derived IL-26 to nucleic acid-triggered innate immune activation and helps amplify inflammatory responses at mucosal and barrier sites. Furthermore, IL-26 produced by CD68^+^ alveolar macrophages and Th17 cells strengthens antimicrobial defense in the lungs by promoting neutrophil recruitment to sites of bacterial infection [21].

Moreover, an increase in neutrophil numbers enhances antiviral protection in the lungs [21,59]. IL-26 supports antiviral immunity through both direct and indirect mechanisms. It has been shown to bind viral RNA and inhibit replication independently of immune cell mediation, as demonstrated in studies on hepatitis C virus (HCV) [92]. Furthermore, IL-26 promotes the expression of TRAIL (TNF-related apoptosis-inducing ligand) on NK cells, enhancing their cytotoxic activity against HCV-infected hepatocytes [17,37]. Due to its highly cationic charge, IL-26 may additionally modulate virus–cell interactions, altering viral infectivity [30]. IL-26 also induces M1 polarization of THP-1-derived macrophages and promotes intracellular killing of *Mycobacterium tuberculosis* through reactive oxygen species (ROS)-dependent mechanisms [93]. Supporting the antiviral role of IL-26 is the finding that its levels are elevated during the acute phase of COVID-19 infection, characterized by hyperinflammation and tissue damage [94,95]. IL-26 is also thought to play a role in the immune response during chronic viral infections. Circulating IL-26 levels are elevated in patients with chronic HCV infection and correlate with the degree of liver inflammation [37]. IL-26 is secreted by circulating CD4^+^ T cells and liver-infiltrating T cells, and it accumulates within hepatocytes despite the lack of detectable IL-26 mRNA in these cells, suggesting that hepatocyte uptake likely occurs from the surrounding tissue [37].

The levels of a comprehensive set of inflammatory cytokines, including IL-26, have been studied in RSV-infected primary human nasal epithelial cells from donors with or without cystic fibrosis. In the medium from the incubated cells at the 72nd hour post-infection, there were no detectable amounts of IL-26, or IL-2, IL-5, IL-13, IL-22, GM-CSF, basic fibroblast growth factor, and macrophage inflammatory protein-1α [96].

However, despite these cellular studies, the available data on IL-26 levels in individuals infected with RSV are lacking. Several studies have quantitatively evaluated various interleukins in RSV bronchiolitis, providing no specific data about IL-26 [97,98,99].

IL-26 protein levels have been significantly elevated in bronchoalveolar lavage samples from patients who have undergone lung transplantation with bronchiolitis obliterans but not with acute rejection [100].

This dual function as both an effector molecule and an immunological amplifier in barrier tissue defense suggests that IL-26 plays a significant role in mucosal immunity, particularly in tissues where its receptor subunit, IL-20RA, is highly expressed, such as the skin, lungs, and intestinal epithelium. Unlike other IL-20 subfamily cytokines like IL-22, which primarily induce antimicrobial peptides through epithelial signaling, IL-26 facilitates communication between epithelial barriers and immune cells [30]. IL-26 physicochemical properties confer functional similarities to cationic antimicrobial peptides such as the human cathelicidin LL-37 [53,91].

IL-26 exhibits a controversial role in mucosal immunity, acting as either a protective or pro-inflammatory mediator depending on the context. This duality could be explained by its two distinct mechanisms of action: receptor-dependent cytokine signaling and receptor-independent antimicrobial and DNA-binding activity. The predominance of one mechanism over the other is influenced by the local tissue environment, the responding cell types, the source of extracellular DNA (microbial versus host-derived), and whether the immune response occurs during acute infection or chronic tissue damage, which may account for apparently contradictory findings across studies.

Several studies support a protective role for IL-26 at mucosal surfaces through its direct antimicrobial activity [21,65,91]. Hawerkamp et al. [78] demonstrated that IL-26 directly kills *Mycobacterium tuberculosis* and reduces infection rates in macrophages, potentially via binding to lipoarabinomannan in the bacterial membrane. In addition, IL-26 stimulation of macrophages and dendritic cells induces the secretion of tumor necrosis factor-α and inflammatory chemokines, contributing to antimicrobial defense [78]. Consistently, IL-26 produced by CD68^+^ alveolar macrophages and Th17 cells has been shown to strengthen pulmonary antimicrobial immunity by promoting neutrophil recruitment to sites of bacterial infection [21].

Through classical cytokine signaling via the IL-20R1/IL-10R2 receptor complex on non-hematopoietic cells, IL-26 also promotes the production of chemokines and antimicrobial peptides that enhance epithelial barrier defenses [21,26].

In contrast, IL-26 can exert pro-inflammatory effects through a receptor-independent mechanism linked to its ability to form complexes with DNA of different origins. The uptake of these complexes in cells activates intracellular nucleic acid sensors such as TLR9, STING, and inflammasome pathways, leading to robust type I interferon and pro-inflammatory cytokine production. This mechanism provides a plausible explanation for the pathogenic role of IL-26 observed in chronic inflammatory conditions such as Crohn’s disease and in settings of sterile inflammation [53]. These findings show that the biological outcome of IL-26 activity is highly cell-type-dependent.

Furthermore, another explanation may lie in the temporal context of IL-26 expression. Transient IL-26 production during acute infection may facilitate microbial clearance, whereas chronic overexpression or sustained tissue damage, accompanied by continuous release of host-derived DNA, may perpetuate a DNA-carrier-driven inflammatory loop characteristic of autoimmune and chronic inflammatory diseases.

## 10. The Role of IL-26 in Inflammatory Skin Diseases

Accumulating evidence indicates that IL-26 exerts proinflammatory, immunomodulatory, and angiogenic effects through multiple cellular and molecular mechanisms, thereby contributing to the development and progression of various skin diseases.

### 10.1. IL-26 in Psoriasis

Psoriasis is a Th17-mediated inflammatory disease. IL-26 expression is markedly increased in lesional skin and contributes to disease progression by promoting angiogenesis and immune cell infiltration [38]. A study with anti-IL-26 monoclonal antibodies has shown suppression of inflammation in mouse models of psoriasis without impairing IL-26’s antimicrobial activity [79].

Single-cell RNA sequencing has revealed IL-26 expression in both CD4^+^ Th17 and CD8^+^ T cells producing IL-17A within psoriatic lesions [101]. Notably, a distinct population of IL-26^+^ Th17 intermediates has been identified. These cells induce transforming growth factor-beta 1 (TGF-β1) expression in keratinocytes via IL-26 receptor signaling and differentiate into IL-17A-producing cells during later stages of Th17 development [15]. IL-26 exerts proinflammatory effects on keratinocytes by upregulating IL-1α and IL-8, thereby facilitating neutrophil recruitment and promoting epidermal hyperplasia [15]. In their study, Baldo and colleagues confirmed that IL-26 enhances the expression of IL-1 family members and IL-8 in keratinocytes. They further demonstrated that IL-26 also upregulates other cytokines, such as TNF-α, and neutrophil-attracting chemokines, including CXCL1 and CXCL8, thus promoting leukocyte recruitment and tissue inflammation in pustular psoriasis [71]. This IL-26-mediated pathway establishes a self-perpetuating cycle that enhances neutrophil mobilization and autoinflammation. Furthermore, IL-26–bacterial DNA complexes trigger TLR9-mediated neutrophil activation, establishing a self-amplifying autoinflammatory loop [21,71]. In pustular psoriasis, neutrophils synthesize and store IL-26. Upon degranulation, neutrophil-derived IL-26 plays a central role in disease exacerbation, challenging the previously held notion that Th17 cells are the primary source of IL-26 in psoriasis [71].

IL-26 also promotes the activation of vascular endothelial cells. A study demonstrated that it enhances the propagation and differentiation of endothelial cells within the human umbilical vein by inducing fibroblast growth factors FGF2 and FGF7. This angiogenic effect is mediated by activation of the PI3K-Akt, MAPK/ERK, and NF-κB pathways [38].

Collectively, these findings indicate that IL-26 contributes to psoriasis through multiple mechanisms, potentially leading to pustular disease manifestations.

### 10.2. IL-26 in Atopic Dermatitis

AD is predominantly a Th2-driven condition but displays features of Th17 involvement, particularly in its acute phase [102,103]. IL-26 mRNA is significantly upregulated in AD lesional skin and correlates positively with Th2-associated genes such as IL-4 and CCL17, as well as with IL-22 and IL-33, though not with IL-17A [45,104]. IL-26 also induces expression of IL-1β, IL-8, IL-33, CCL20, and β-defensin 2 in normal human epidermal keratinocytes (NHEKs) via activation of STAT1 and STAT3, through JAK1, JAK2, and TYK2 signaling [7,14]. A recent study showed that IL-26 has limited proinflammatory activity alone but, together with IL-1β, amplifies the epidermal secretion of CXCL1, CCL20, and TSLP via Jak/STAT signaling [105].

Research on serum levels of IL-26 in AD is scarce. Kamijo and colleagues found that the serum IL-26 levels in AD individuals, regardless of disease severity, were similar to those in healthy controls [45]. However, another study demonstrated the significant upregulation of IL26 mRNA in AD skin lesions [105].

IL-26 localized action in the skin may facilitate the crosstalk between the Th2 and Th17 axes. IL-19, which is induced by IL-17A and promotes Th2 cytokine production, may represent a mechanistic bridge between these pathways [106]. These findings implicate IL-26 as a modulator of immune polarization in AD.

### 10.3. IL-26 in Allergic Contact Dermatitis

In patients with allergic contact dermatitis (ACD), IL-26 expression is significantly upregulated in the skin (with or without lesions), as well as in peripheral blood mononuclear cells (PBMCs). Plasma IL-26 levels are also elevated relative to healthy controls [107]. These findings suggest that IL-26 may contribute to T-cell-mediated cytotoxic mechanisms characteristic of ACD.

### 10.4. IL-26 in Bullous Pemphigoid

Bullous pemphigoid is an autoimmune blistering disease. There is hardly any data on the involvement of IL-26 in the pathophysiology of this disease. It was shown by Mizuno and colleagues that IL-26 induces infiltrating neutrophils or eosinophils in this context [49]. Elevated IL-26 levels are found in patient serum and lesional skin, where IL-26 forms complexes with extracellular DNA. These IL-26-DNA complexes promote inflammation by inducing IL-1β and IL-6 production in monocytes and neutrophils and by enhancing the expression of proteases, such as neutrophil elastase and MMP-9, which contribute to dermal–epidermal separation [49].

## 11. IL-26 in Rheumatoid Arthritis and Other Inflammatory Arthritides

Th17 cells are essential for the pathogenesis of RA [108,109] and osteoarthritis [81]. IL-26 has emerged as an early and potent contributor to disease initiation. Elevated IL-26 levels have been detected in the serum and synovial fluid (SF) of patients with RA, with notably higher concentrations in SF, suggesting local production within inflamed joints [14]. Immunohistological analysis of synovial tissue revealed that synoviocytes are a major source of IL-26. Functionally, synovial-derived IL-26 activates myeloid lineage cells, including monocytes, dendritic cells, and macrophages, inducing robust IL-1β production and promoting M1 macrophage polarization and osteoclast differentiation [14]. Interestingly, monocytes and NK cells, which express little to no IL-26 receptor, remain responsive to IL-26, suggesting the involvement of receptor-independent signaling mechanisms [34]. However, the precise molecular mechanisms remain to be elucidated. IL-26 levels in RA patients are positively correlated with IL-1β concentrations in both serum and SF, but are not significantly associated with C-reactive protein, IL-6, TNF-α, or leukocyte counts [14]. These findings underscore IL-26’s distinct contribution to RA pathophysiology, particularly through IL-1β-driven inflammatory pathways. Moreover, in chondrocytes, IL-26 exerts pro-inflammatory effects by enhancing the expression of cyclooxygenase-2 (COX-2), IL-6, and MMP in a MAPK-dependent manner. IL-26 downregulates Col II in chondrocytes and promotes cartilage degeneration [80]. IL-26 has been shown to activate JAK-STAT, MAPK, and NF-κB signaling to increase the expression of IL-20R1 and receptor activator of nuclear factor-kappaB ligand (RANKL) [81]. RANKL, produced by T cells, synovial fibroblasts, and stromal cells, plays a central role in promoting osteoclastogenesis and bone resorption by interacting with the RANK receptor on osteoclast precursors. Lee and colleagues were the first to demonstrate that IL-26 promotes osteoclastogenesis by both enhancing RANKL expression in FLSs and directly stimulating osteoclast differentiation. These findings suggest the potential role of the IL-26-IL-20RA-RANKL axis as a prospective therapeutic target in RA [81].

Beyond RA, elevated IL-26 concentrations have also been observed in the synovial fluid of patients with spondyloarthritis, supporting a broader role for IL-26 in inflammatory joint diseases [72,110].

## 12. IL-26 in Inflammatory Bowel Disease

IBD, which includes CD and UC, is associated with chronic gut inflammation driven by immune dysregulation. Th17 cells are central to IBD pathogenesis, and increasing evidence indicates that IL-26 contributes to intestinal inflammation [111]. Notably, the number of IL-26-producing cells was significantly elevated in active CD lesions [55]. Double immunofluorescence identified CD4^+^ T cells, NK cells, and macrophages as the primary cellular sources of IL-26 in this disorder, in addition to RORγt^+^ Th17 cells and colonic SEMFs [16,55,111]. Patients with IBD exhibit elevated IL-26 gene expression in inflamed colonic mucosa [16,53], which correlates significantly with increased IL-22 and IL-8 expression in CD. This elevated local expression of IL-26 in the colonic mucosa is also mirrored by increased serum levels in patients with CD compared to healthy controls [55]. Other studies confirmed that in the gastrointestinal tract IL-26 induces IL-8 production by epithelial cells and subepithelial myofibroblasts, both of which express IL-20R1 and IL-10R2, and further demonstrated that IL-26 stimulates these cells to produce IL-6 and TNF-α [16,55]. Mechanistically, IL-26 activates STAT1 and STAT3, as well as several mitogen-activated protein kinases (MAPKs), including p42/44, SAPK/JNK, and p38 MAPK, and the PI3K/Akt pathway. These signaling cascades contribute to the proinflammatory effects of IL-26 in colonic SEMFs. Moreover, IL-26 stimulation induces nuclear translocation of NF-κB and phosphorylated c-Jun (AP-1), implicating these transcription factors in IL-26-mediated cytokine gene expression [16,77].

A genetic association study further implicated IL-26 in IBD, identifying the rs2870946 polymorphism as a potential risk factor [53]. IL-26 expression correlates with Th17-related gene signatures and disease severity in IBD [55]. In individuals with common variable immunodeficiency and inflammatory complications, IL-22^+^IL-26^+^ NK22 cells infiltrate mucosal tissues, including the gastrointestinal tract [53]. Furthermore, polymorphisms in the IL26 gene may influence immune responses in IBD. Pinero and colleagues reported that PBMCs from CD patients carrying a variant IL26 genotype (varIL26) exhibited reduced bacterial-killing capacity and lower serum IL-26 levels compared to wild-type IL26 (wtIL26) carriers [112]. Despite similar levels of circulating bacterial DNA, varIL26 carriers displayed elevated IFN-γ, IL-12, and TNF-α levels, suggesting impaired bacterial clearance and exacerbated inflammatory responses due to altered IL-26 function [112]. Collectively, these findings support a pathogenic role for IL-26 in IBD, particularly in CD, where it contributes to local inflammation through cytokine induction and modulation of both epithelial and immune cell function.

## 13. IL-26 in Asthma

Growing evidence suggests an association between IL-26 and asthma. Studies indicate that IL-26, detected in the sputum of asthmatic patients, promotes Th17 cell differentiation and stimulates local monocytes and macrophages in the airways to produce proinflammatory cytokines [90]. IL-26 has been proposed as a potential biomarker for asthma severity, particularly in patients with non-Th2-dominant inflammation [113]. In children with uncontrolled asthma, the levels of IL-26 in induced sputum were higher than those in children with controlled disease [113]. Additionally, other researchers reported that systemic IL-26 concentrations are elevated in adult asthma patients relative to non-asthmatic controls, independent of disease severity or atopic status [114]. Avramenko and colleagues [115] reported that IL-26 levels in exhaled breath could reliably distinguish asthmatic patients from healthy individuals. They found positive correlations between serum and exhaled cytokine levels, including a link between eosinophil count and serum IL-26. Notably, obese individuals with asthma exhibited significantly higher systemic IL-26 concentrations compared to healthy controls, whereas non-obese asthmatic patients did not show a clear increase in systemic IL-26 relative to non-asthmatic controls [116]. In addition to confirming elevated serum IL-26 levels, Louhaichi and colleagues demonstrated higher IL-26 expression at the mRNA level in PBMCs from severe asthmatic patients compared with healthy controls [90]. Recombinant human IL-26 (rIL-26)-stimulated cell culture supernatants from patients contained elevated levels of IL-17A, IL-6, IL-1β, and TNF-α compared with those from healthy controls. In contrast, rIL-26 did not affect the production of Th2- or Treg-associated cytokines by CD4^+^ T cells cocultured with autologous monocytes in either asthmatic patients or healthy controls. Notably, rIL-26-stimulated monocytes and memory CD4^+^ T cells promoted Th17 differentiation. Moreover, rIL-26 induced the expression of IL-17A and the Th17 lineage-defining transcription factor RORγt in memory CD4^+^ T cells [90]. These observations align with results reported in COPD, where elevated IL-26 levels have been associated with systemic inflammation, reduced lung function, and increased body weight [116].

Epidemiological data from large patient cohorts have associated COVID-19 infection in children with the development of allergic predisposition [117]. Significant downregulation of ACE2, which serves as a coronavirus receptor, has been observed in nasal and airway epithelial cells of patients with type 2 asthma and allergic rhinitis. Furthermore, the expression levels of ACE2 have been negatively associated with type 2 cytokines. As a result, ACE2 downregulation may impede viral infiltration in host cells, but at the same time may increase type 2 inflammatory response [118].

According to the current mechanistic model, virus-induced epithelial damage causes a surge in alarmin cytokines (IL-25, IL-33, and TSLP), which stimulate the differentiation of naive CD4+ T cells into Th2 helpers and the expansion of group 2 innate lymphoid cells. It also promotes long-lasting epigenetic changes and reprogramming of hematopoietic stem cells towards the Th2 phenotype. IL-7 and IL-15 alter T- and B-cell homeostasis through their roles in immune memory, while impaired T regulatory cell function may diminish immune tolerance [117]. In allergic airways, IL-33 often modulates Th2-mediated repair, promoting TGF-β and collagen accumulation, whereas in the non-allergic respiratory system, IL-33 may exacerbate the IFN-γ-driven cytokine storm [119]. Given the role of IFN-γ in that cytokine storm, it is noteworthy to mention that the IL-26 gene is located in close proximity to the IFN-γ gene and even shares an enhancer sequence with it [53]. Provided that IL-26 expression significantly correlates with the severity of COVID-19 infection [94,95], the potential functional involvement of IL-26 in COVID-19 can be suspected.

## 14. Other Inflammatory Diseases

IL-26 has also been associated with other inflammatory diseases, such as SLE [48], hidradenitis suppurativa [120], and Behçet’s disease (BD) [76]. Individuals with BD have elevated levels of IL-26 in their blood compared to healthy individuals, and these levels are linked to the severity of their symptoms. Researchers found a strong connection between the levels of IL-26, IL-17, and IL-37 in the blood of these patients [76].

## 15. IL-26 in Cancer Development

In recent years, a growing body of research has investigated the role of IL-26 in the initiation and promotion of malignant transformation, as well as in tumor invasion and metastasis. Given the well-documented role of IL-26 in inflammatory processes, a rationale for investigating such an association is provided by the substantial link between inflammation and tumorigenesis [121]. Elevated serum levels of IL-26 have been implicated in gastric cancer [122], in which higher serum IL-26 strongly correlates with disease stage. In hepatocellular carcinoma (HCC), elevated IL-26 expression in resected histological specimens may serve as a prognostic marker of unfavorable disease onset [123]. Meanwhile, experimental evidence has suggested that IL-26 is mechanistically involved in TNBC resistance to EGFR-tyrosine kinase inhibitors (EGFR-TKI) [39]. An in vivo investigation identified IL-26 as a pivotal mediator influencing the engraftment and metastatic progression of TNBC [82]. Drawing upon experimental findings, the authors determined that the impact of IL-26 on engraftment and metastasis was predominantly contingent upon neutrophil activity. Furthermore, it was demonstrated that IL-26 plays a significant role in the induction of inflammatory cytokines in response to DNA. The authors also reported increased IL26 transcript levels in both malignant cells and Th17 CD4^+^ T lymphocytes within the analyzed TNBC tissue specimens [82]. Furthermore, upregulation of IL-26 in malignant pleural effusion promotes progression of the condition through CD4+IL-22+ T-cell differentiation and CD8^+^ T-cell suppression [124]. In all these examples, elevated IL-26 levels are directly associated with cancer phase or progression. A possible explanation for this association is provided by Wang and colleagues, who demonstrated that stimulation of gastric cancer cells with IL-26 significantly enhanced cell proliferation. Moreover, co-stimulation with IL-17A and IL-26 produced an even stronger proliferative effect. The same study demonstrated that, in gastric cancer, tumor cells may recruit Tc17 cells via a mechanism involving the CXCL16–CXCR6 axis, as confirmed by spatial transcriptomics. Subsequently, the Tc17-derived IL-17A and IL-26 further promoted tumor progression, as validated by CCK-8, wound-healing, and transwell assays. These findings indicate that IL-17A and IL-26 not only independently promote malignant phenotypes of gastric cancer cells but also act synergistically to accelerate tumor progression [125].

In contrast, increased IL-26 expression levels in colorectal cancer were related to good prognosis [33]. A comparable relationship has also been demonstrated in a recent study employing multiple bioinformatics approaches, which revealed that IL-26 transcriptional levels are significantly elevated in renal cell carcinoma and that such elevations are associated with an improved clinical outcome [126].

Intriguingly, in colon cancer, Wei and colleagues did not find significant expression in cancer samples compared to normal tissues [127].

The available data suggest that IL-26 may serve as a potential biomarker for specific malignant conditions. To date, investigations into IL-26 as a biomarker have been conducted in breast cancer [82], colorectal cancer [33], pancreatic ductal adenocarcinoma [128], and HCC [123].

Although emerging evidence suggests a possible involvement of IL-26 in tumorigenesis, further research efforts are required to clarify its role, including its potential utility as a biomarker for specific malignancies or as a therapeutic target.

Downregulation of IL-26 has been observed in oral squamous cell carcinoma and has been related to poor prognosis [129]. Although IL-26 may enhance type I interferon secretion and presumably activate dormant cancer stem cells, its reduced expression may limit drug efficacy [130] and exacerbate survival outcomes.

The contradictory role of IL-26 in cancer may be partially explained by its essential role in autophagy. Increased levels of IL-26 in HCC have been correlated with both unfavorable and favorable prognoses [123,131]. In general, autophagy is believed to suppress cancer development at an early stage. However, during cancer progression, the elevated autophagic activity often promotes cancer survival and proliferation [131]. Consistent with this assumption, both stimulation and suppression of autophagy have been established as therapeutic approaches for emerging and advanced malignancies, respectively [132,133]. Furthermore, IL-26 protein levels in HCC strongly depend on the disease stage [131]. More importantly, in HCC, IL-26 has been mechanistically implicated in autophagy, presumably via a JAK/STAT3-dependent mechanism. In HCC cell lines, IL-26 treatment increases the number of autophagic vacuoles and LC3-II expression, both of which are crucial for autophagosome formation. In contrast, the opposite tendency has been observed for SQSTM1/p62 and Bcl2 following IL-26 treatment [131]. The tumor microenvironment (TME) significantly modulates the impact of IL-26 on tumor formation and progression. Extensive evidence supports the overexpression of IL-26 in inflammatory contexts and the contribution of IL-26-induced cytokines, including IL-6, IL-8, and CXCL1, to TNBC metastasis [39]. In TME, exposure to IL-26 triggers an EGFR-TKI bypass via AKT- and JNK-mediated signaling to promote TKI resistance. Moreover, the influence of IL-26 on EphA3 and EGFR signaling suppresses ER stress pathways in TNBC [39]. The markedly higher IL-26 levels in HCC patients at least partially originate from macrophages residing in the tumor mass, as suggested by a significant overlap between IL-26 and CD68 immunostaining [131]. In general, inflammation triggered by infiltrating innate immune cells contributes to tumor growth and metastasis [82,134]. Upon entering the TME and subsequent activation, infiltrating immune cells amplify inflammatory signaling, thereby directly contributing to tumor proliferation and metastasis [82,134,135,136,137,138]. In this respect, TNBC tissues are often markedly infiltrated by immune cells, particularly T cells, neutrophils, and macrophages [82,139,140,141]. Consistent with the previous findings, an elevated number of TNBC-infiltrating CD8^+^ T cells may indicate a favorable prognosis and chemotherapy sensitivity. In contrast, the presence of tumor-infiltrating macrophages has been associated with poor disease outcome [141].

IL-26 suppresses the growth of intestinal epithelial cells [53,55], although it promotes the proliferation of gastric tumor cells [53,142]. The primary cellular source of IL-26, Th17 cells, is evidently linked to gastric cancer [122,143]. The presence of infiltrating Th17 cells in malignant gastric tissues may result from differentiation of CD4^+^ T cells or from migration of Th17 cells from the bloodstream into tumor tissues under the influence of the TME [122,144,145]. Th17 cells are known for their complex roles in various malignancies, facilitating tumor progression through advancing angiogenesis and immune suppression. Meanwhile, Th17 may also promote antitumor immune responses by enhancing the activity of CD8^+^ T cells [146]. In tumors, Th17 cells exhibit remarkable plasticity, acquiring phenotypes such as Th1 cells or regulatory T cells in response to the TME. Their plasticity is essential for their function within the TME, which can modulate Th17 activity through cytokines, engagement of signaling cascades, and epigenetic mechanisms [146]. A recent study of patients with colorectal cancer found significantly higher IL-26 RNA and protein levels in the non-metastatic group than in the metastatic group. Additionally, the IL-26 high-expression group demonstrated improved overall and disease-free survival. Moreover, the pool of differentially expressed genes positively correlating with high IL-26 levels was enriched for JAK/STAT and epsilon receptor signaling [33]. Interleukin-mediated JAK/STAT activation, subsequently causing anti-apoptotic but not anti-inflammatory or proliferative effects in primary colonic epithelial cells, has been previously described for another interleukin, IL-11 [147]. Taken together, the existence of such non-metastatic and non-proliferative associations of IL-26 and JAK/STAT signaling further complicates their potential role in cancer progression, cellular proliferation, and apoptosis in the context of the TME.

## 16. Conclusions

IL-26 is a multifunctional cytokine whose role in immunity and disease is yet to be fully elucidated. Despite significant progress in defining the contribution of IL-26 across various pathological contexts, many aspects of its involvement in regulating immune responses to both external and internal stimuli remain unclear. For instance, the exact spatiotemporal patterns of IL-26 expression in both normal and pathological conditions are not fully understood. On the other hand, the ability of IL-26 receptor subunits to associate with other cytokines creates new opportunities to investigate the detailed structural mechanisms of these interactions. Moreover, one of the cytokines with which the subunits of the IL-26 receptor associate is another cytokine of the IL-10 family, IL-20 [148]. IL-20 elicits responses in its target cells through two types of membrane receptors, IL-20R1/IL-20R2 and IL-22R1/IL-20R2. Whether such alternative receptors also exist for IL-26 is yet to be determined. The context-dependent duality of IL-26 constitutes a key unresolved question, as the mechanisms controlling the balance between its beneficial and deleterious effects, such as the tissue microenvironment, receptor expression, and immune cell composition, remain incompletely understood.

Further research efforts will determine whether IL-26 may serve as a biomarker. IL-26 levels have been shown to correlate with disease activity in several inflammatory conditions, suggesting its potential utility as a biomarker of Th17-driven inflammation and possibly as a predictor of response to biologic therapies. Experimental and methodological limitations include the absence of IL-26 in rodents, which limits the use of conventional mouse models. The expanded application of single-cell and multi-omics approaches to identify IL-26-responsive cell populations could overcome these limitations and will be critical for advancing both mechanistic understanding and the translational potential of IL-26 biology. Crucial aspects that require further validation include whether DNA-binding is an essential physiological function of IL-26 or part of a pathological amplification loop.

Addressing these questions, together with antimicrobial and antiviral properties of IL-26 and its role in the development of diverse inflammatory and autoimmune diseases, will further establish IL-26 as a cytokine of considerable scientific interest and a potential therapeutic target for a range of disorders.

## Figures and Tables

**Figure 1 ijms-27-00325-f001:**
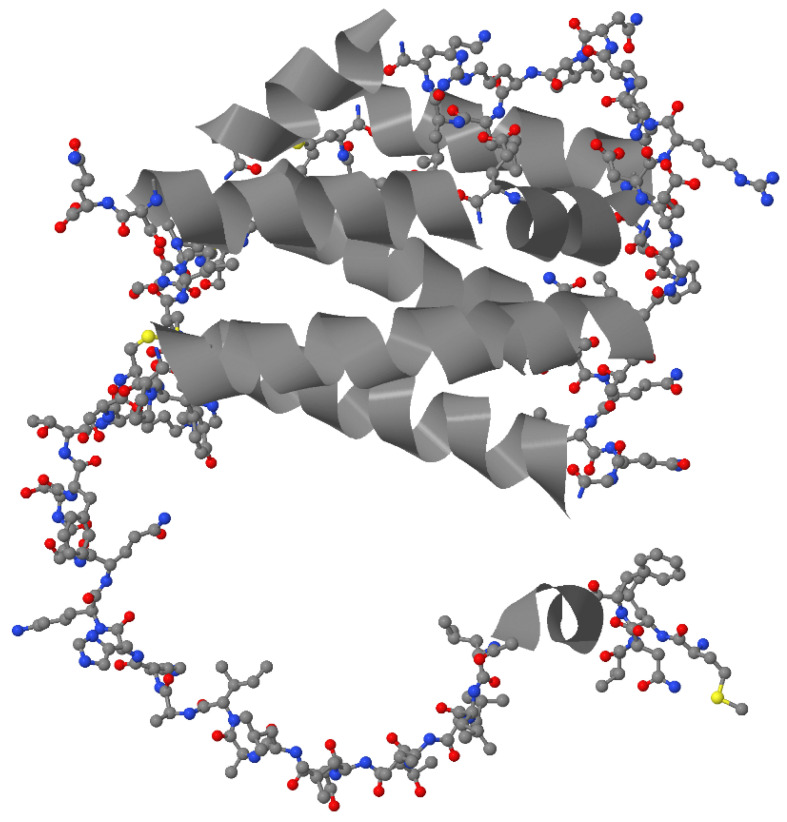
A model of the molecular structure of IL-26 [UniProtKB entry Q9NPH9]. α-helices are depicted as grey ribbons, while all other regions within the molecule are presented as a ball-and-stick model (carbon, nitrogen, oxygen, and sulfur atoms are depicted as grey, blue, red, and yellow balls, respectively).

**Figure 2 ijms-27-00325-f002:**
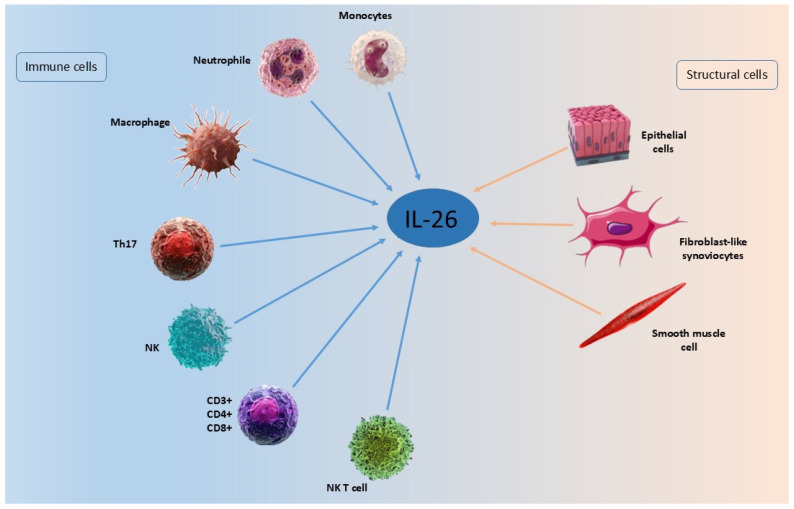
Cellular sources of IL-26.

**Figure 4 ijms-27-00325-f004:**
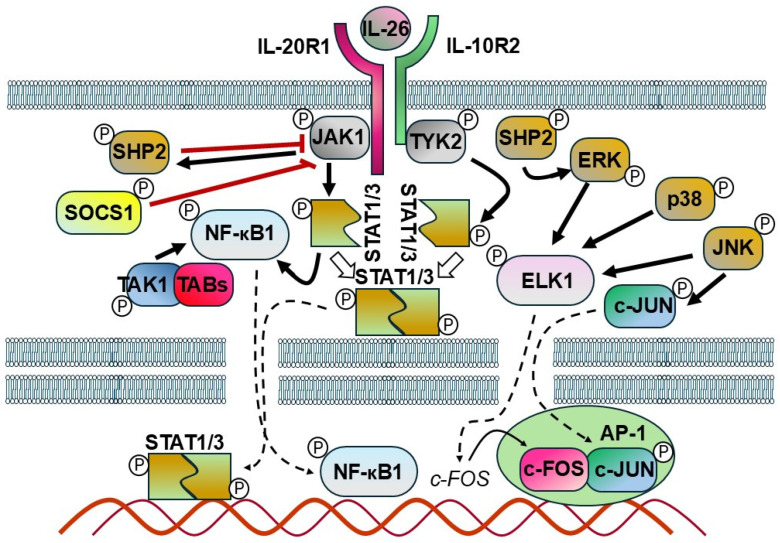
Schematic model representing the potential mechanism of IL-26 signaling. The interaction between IL-26 and its receptor recruits and activates Janus kinases (JAKs). These activated JAKs, in turn, phosphorylate STAT proteins, which are then translocated as dimers into the nucleus where they regulate the expression of various genes. In response to IL–receptor interactions, SHP2 is phosphorylated by JAKs and serves as an early feedback inhibitor. In addition, SHP2 plays a central role in the activation of ERK downstream of various growth factor receptors. Another inhibitory factor, SOCS1, directly suppresses JAK1, JAK2, and TYK2 activity. TAK1 triggers the activation of the sub IKK complex, which subsequently results in the degradation of IκBα. The released NF-κB is activated and translocated into the nucleus, where it modulates gene transcription. JNK mitogen-activated protein kinase (MAPK) directly phosphorylates and activates c-Jun. JNK, p38, and ERK phosphorylate and activate ELK-1, which promotes the upregulation of c-Fos. C-Jun and c-Fos form a dimer termed the AP1 complex, which regulates the expression of multiple target genes (bold black arrows: activation, white arrows: dimerization, bold red lines: suppression/inhibition, dashed arrows: transfer to the nucleus, thin black arrow: gene expression).

**Table 1 ijms-27-00325-t001:** IL-20 subfamily cytokines and their respective receptors.

IL-20 Subfamily Cytokine	Receptor Complex	Key References
IL-19	IL-20R1/IL-20R2	[8,9]
IL-20	IL-20R1/IL-20R2; IL-22R1/IL-20R2	[9,10]
IL-22	IL-22R1/IL-10R2	[11,12]
IL-24	IL-20R1/IL-20R2; IL-22R1/IL-20R2	[9,13]
IL-26	IL-20R1/IL-10R2	[6,7]

**Table 2 ijms-27-00325-t002:** IL-26-producing cell populations across various tissues.

IL-26 Cell Source	Tissue	Condition	References
Monocytes	Peripheral blood mononuclear cells (PBMCs)	*Mycobacterium tuberculosis* in monocytes	[67]
Neutrophils	Lesional skin biopsies	Psoriasis	[71]
Macrophages	Blood samples; IBD mucosa	RA IBD	[14,16]
Th17	Inflamed colonic lesions; Lesional human psoriatic skin	CD Psoriasis	[50,55]
NK	IBD mucosa; mucosa-associated lymphoid tissues; PBMCs	IBD; secondary lymphoid tissue	[16,34,62]
NK T cells	Mononuclear cell isolates	Tuberculous pleurisy	[64]
CD3^+^, CD4^+^, CD8^+^	Serum; skin section	Bullous pemphigoid; ulcerative colitis	[16,49,53,54,59]
B cells	Cell lines	Human herpesvirus-infected B cells	[17,36]
Primary bronchial epithelial cells	Bronchial brush biopsy	Stimulation with TLR3 agonist poly-IC	[56]
FLS	FLS cell line	RA	[14]
Smooth muscle cells	Smooth muscle cells	AAV; spondyloarthritis	[25,72]
Myofibroblast	Colonic myofibroblasts; smooth muscle actin-expressing myofibroblasts	IBD; spondyloarthritis	[16,72]

## Data Availability

No new data were created or analyzed in this study. Data sharing is not applicable to this article.

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
