# Peer review of "Multifaceted Roles of IL-26 in Physiological and Pathological Conditions"

_ijms, 2025, doi:10.3390/ijms27010325_

Round 1
Reviewer 1 Report
Comments and Suggestions for Authors
This manuscript presents a comprehensive and well-structured review on IL-26, covering its biological properties, receptor biology, DNA-binding behavior, and involvement in immunity, inflammation, and cancer. The topic is timely and relevant to the readership of IJMS. The manuscript is generally well written and supported by a large body of literature. It was also submitted to the proper Special Issue.
However, several issues need attention before the manuscript can be considered for publication. The most important concerns are presented below.
Detailed comments:
The iThenticate score (currently 47%) must be decreased, preferably below 30%.
The authors state “Despite being a relatively recent discovery” – please provide some short history of discovery and works on IL-26.
Line 45, the authors should cite https://doi.org/10.56782/pps.264 as example
The description of the IL-20 cytokine family would benefit from a schematic or table to support comprehension.
Line 54, the authors should mention IL-23 and cite https://doi.org/10.56782/pps.262
Figure 1, the authors should state whether (and how many) structures of Il-26 are present in the databases such as RCSB or AlphaFold.
The authors state that naturally secreted IL-26 shows stronger antibacterial activity than recombinant IL-26. Please explain why (glycosylation? cofactors? synergy?). Currently, this point is presented without explanation.
Were the figures created using BioRender or any other tool like this one? If yes, it should be clearly stated.
Line 475, why is “Rheumatiod” with capital “R”?
The antiviral section mentions HCV and COVID-19 but not recent findings on IL-26 in respiratory viral infections (RSV, influenza). Why?
IBD Consider adding a subsection summarizing contradictory findings or gaps (e.g., why IL-26 seems protective in some mucosal contexts but pathogenic in others).
The authors could also mention the role of in-silico methods in the studies on IL-26.
The conclusion is clear but could also include some future perspectives, therapeutic perspectives, unanswered questions, hypothesis that need to be verified.
Author Response
Comment 1: The iThenticate score (currently 47%) must be decreased, preferably below 30%.
Response 1: In order to meet the requirements established by the journal and pointed out by the reviewer, we officially requested the original iThenticate report from the editorial team of IJMS. In it, we noticed that most similarities arise from the reference list, which we properly formatted. We also rephrased numerous expressions throughout the manuscript.
Comment 2: The authors state “Despite being a relatively recent discovery” – please provide some short history of discovery and works on IL-26.
Response 2: The following explanatory text was inserted in the main text of the manuscript (lines ).
IL-26 was first identified by Knappe et al. (2000), who used subtractive hybridization to characterize a novel cDNA, termed ak155. It was specifically expressed in human T cells transformed by subgroup C strains of herpesvirus saimiri (HVS). The new transcript showed weak nucleotide homology to the cellular interleukin-10 (IL-10) gene. Northern blotting and RT-PCR analysis revealed strong expression of ak155 in HVS-transformed human T-cell lines and low-level expression in unstimulated peripheral blood cells from healthy humans. Initial functional analyses showed that recombinant AK155 forms dimers, both when expressed in Escherichia coli and in eukaryotic cells. The gene mapped to chromosome 12q15 at a locus later defined as IL-26.
Comment 3: Line 45, the authors should cite https://doi.org/10.56782/pps.264 as example
Response 3: The required reference was included as an example in the manuscript (Reference 4 in the reference list).
Kuśnierek, M.; Czerwińska, M. Lycium barbarum polysaccharide fraction—Isolation from fruits and impact on the secretion of inflammatory mediators by human mononuclear cells and neutrophils. Prospects Pharm. Sci. 2024, 22, 86–92. https://doi.org/10.56782/pps.264
Comment 4: The description of the IL-20 cytokine family would benefit from a schematic or table to support comprehension.
Response 4: We inserted the required table that presents the cytokines of the IL-20 subfamily and their respective receptors. References are provided in the manuscript.
Table 1. IL-20 subfamily cytokines and their respective receptors.
|
IL-20 subfamily cytokine |
Receptor complex |
Key references |
|
IL-19 |
IL-20R1 / IL-20R2 |
[8, 9] |
|
IL-20 |
IL-20R1 / IL-20R2; IL-22R1 / IL-20R2 |
[9, 10] |
|
IL-22 |
IL-22R1 / IL-10R2 |
[11, 12] |
|
IL-24 |
IL-20R1 / IL-20R2; IL-22R1 / IL-20R2 |
[9, 13] |
|
IL-26 |
IL-20R1 / IL-10R2 |
[6, 7] |
Comment 5: Line 54, the authors should mention IL-23 and cite https://doi.org/10.56782/pps.262
Response 5: The required reference was included as an example in the manuscript (Reference 5 in the reference list).
Olszewski, J.; Kozon, K.; Sitnik, M.; Herjan, K.; Mikołap, K.; Gastoł Bara, M.; Armański, P.; Sawczuk, M. Mirikizumab – A New Option in the Treatment of Inflammatory Bowel Diseases. Prospects Pharm. Sci. 2024, 22, 178–185. https://doi.org/10.56782/pps.262
Comment 6: Figure 1, the authors should state whether (and how many) structures of Il-26 are present in the databases such as RCSB or AlphaFold.
Response 6: As a result from a rigorous, detailed examination of the RCSB or AlphaFold databases, we identified a single structure for IL-26, the Q9NPH9 entry mentioned in the manuscript.
Comment 7: The authors state that naturally secreted IL-26 shows stronger antibacterial activity than recombinant IL-26. Please explain why (glycosylation? cofactors? synergy?). Currently, this point is presented without explanation.
Response 7: In the manuscript, this is explained as follows:
No post-translational modifications of IL-26 have been conclusively characterized to date. However, differences in activity have been observed between naturally produced and recombinant forms of IL-26. For instance, IL-26 secreted by Th17 cells shows greater antibacterial activity than recombinant IL-26 produced in prokaryotic systems, suggesting that post-translational modifications or synergistic interactions with other immune molecules may enhance its function [21, 30].
Sources:
Meller, S.; Di Domizio, J.; Voo, K.S.; Friedrich, H.C.; Chamilos, G.; Ganguly, D.; Conrad, C.; Gregorio, J.; Le Roy, D.; Roger, T.; Ladbury, J.E.; Homey, B.; Watowich, S.; Modlin, R.L.; Kontoyiannis, D.P.; Liu, Y.J.; Arold, S.T.; Gilliet, M. T(H)17 Cells Promote Microbial Killing and Innate Immune Sensing of DNA via Interleukin 26. Nat. Immunol. 2015, 16, 970-979. https://doi.org/10.1038/ni.3211
Stephen-Victor, E.; Fickenscher, H.; Bayry, J. IL-26: An Emerging Proinflammatory Member of the IL-10 Cytokine Family with Multifaceted Actions in Antiviral, Antimicrobial, and Autoimmune Responses. PLoS Pathog. 2016, 12, e1005624. https://doi.org/10.1371/journal.ppat.1005624
Comment 8: Were the figures created using BioRender or any other tool like this one? If yes, it should be clearly stated.
Response 8: No, we haven’t used BioRender or any other AI tool producing schematic visualizations. All figures were drawn using the commercially available software MS PowerPoint.
Comment 9: Line 475, why is “Rheumatiod” with capital “R”?
Response 9: According to the official medical standard accepted in Bulgaria, diagnoses should be written with the first letter of the first word capitalized. We understand that this is indeed unnecessary in the case of manuscript writing, where the names of conditions are mentioned in general. Thus, we corrected the misspelled disease name.
Comment 10: The antiviral section mentions HCV and COVID-19 but not recent findings on IL-26 in respiratory viral infections (RSV, influenza). Why?
Response 10: We have expanded section 9. IL-26 as an antimicrobial and antiviral cytokine of the revised manuscript to address IL-26 in the context of respiratory viral infections. We now note that IL-26 was not detected in RSV-infected primary human nasal epithelial cells and that, despite extensive cytokine profiling in RSV bronchiolitis, clinical data on IL-26 levels in RSV-infected patients are currently lacking, highlighting an important knowledge gap. We also include evidence of elevated IL-26 levels in bronchoalveolar lavage fluid from lung transplant recipients with bronchiolitis obliterans, a condition linked to inflammatory and viral airway injury. These additions address the Reviewer’s concern and underscore the need for further studies on IL-26 in respiratory viral infections The following text was included in the manuscript with the provided references renumbered accordingly (lines 490 - 500).
The levels of a comprehensive set of inflammatory cytokines, including IL-26, have been studied in RSV-infected primary human nasal epithelial cells from donors with or without cystic fibrosis. In the medium from the incubated cells at the 72nd hour post-infection, there were no detectable amounts of IL-26, as well as IL-2, IL-5, IL-13, IL-22, GM-CSF, basic fibroblast growth factor, and macrophage inflammatory protein-1α [1].
However, despite these cellular studies, the available data on IL-26 levels in individuals infected with RSV are lacking. Several studies have quantitatively evaluated various interleukins in RSV bronchiolitis, providing no specific data about IL-26 [2-4].
IL-26 protein levels have been significantly elevated in bronchoalveolar lavage samples from patients who have undergone lung transplantation with bronchiolitis obliterans but not with acute rejection [5].
[1] Duan, W.; Cen, Y.; Lin, C.; Ouyang, H.; Du, K.; Kumar, A.; Wang, B.; Avolio, J.; Grasemann, H.; Moraes, T.J. Inflammatory epithelial cytokines after in vitro respiratory syncytial viral infection are associated with reduced lung function. ERJ Open Res. 2021, 7, 00365-2021. https://doi.org/10.1183/23120541.00365-2021
[2] Lee, C.Y.; Sung, C.H.; Wu, M.C.; Chang, Y.C.; Chang, J.C.; Fang, Y.P.; Wang, N.M.; Chou, T.Y.; Chan, Y.J. Clinical characteristics and differential cytokine expression in hospitalized Taiwanese children with respiratory syncytial virus and rhinovirus bronchiolitis. J. Microbiol. Immunol. Infect. 2023, 56, 282–291. https://doi.org/10.1016/j.jmii.2022.08.013
[3] Kostadinova, E.; Angelova, S.; Tsonkova-Popova, T.; Zlateva, D.; Yordanova, R.; Stanilova, S. Systemic IL-10 and IFN-γ levels in respiratory syncytial virus- and rhinovirus-infected Bulgarian children with acute bronchiolitis and their impact on clinical manifestation. Pathogens 2025, 14, 426. https://doi.org/10.3390/pathogens14050426
[4] Barnes, M.V.C.; Openshaw, P.J.M.; Thwaites, R.S. Mucosal immune responses to respiratory syncytial virus. Cells 2022, 11, 1153. https://doi.org/10.3390/cells11071153
[5] Magnusson, J.M.; Ericson, P.; Tengvall, S.; Stockfelt, M.; Brundin, B.; Lindén, A.; Riise, G.C. Involvement of IL-26 in bronchiolitis obliterans syndrome but not in acute rejection after lung transplantation. Respir. Res. 2022, 23, 108. https://doi.org/10.1186/s12931-022-02036-3
Comment 11: IBD Consider adding a subsection summarizing contradictory findings or gaps (e.g., why IL-26 seems protective in some mucosal contexts but pathogenic in others).
Response 11: We have added a dedicated subsection to the revised manuscript (lines 509 -541) that addresses the contradictions and knowledge gaps regarding the role of IL-26 in mucosal immunity. This new section summarizes evidence supporting both the protective and pathogenic functions of IL-26 and discusses potential mechanisms underlying these divergent outcomes. Specifically, we highlight the context-dependent balance between receptor-dependent cytokine signaling and receptor-independent antimicrobial and DNA-binding activities, as well as the influence of tissue microenvironment, responding cell types, the origin of extracellular DNA, and the temporal dynamics of IL-26 expression during acute versus chronic inflammation.
IL-26 exhibits a controversial role in mucosal immunity, acting as either a protective or pro-inflammatory mediator depending on the context. This duality could be explained by its two distinct mechanisms of action: receptor-dependent cytokine signaling and receptor-independent antimicrobial and DNA-binding activity. The predominance of one mechanism over the other is influenced by the local tissue environment, the responding cell types, the source of extracellular DNA (microbial versus host-derived), and whether the immune response occurs during acute infection or chronic tissue damage, which may account for apparently contradictory findings across studies.
Several studies support a protective role for IL-26 at mucosal surfaces through its direct antimicrobial activity [21, 56, 90]. Hawerkamp et al. demonstrated that IL-26 directly kills Mycobacterium tuberculosis and reduces infection rates in macrophages, potentially via binding to lipoarabinomannan in the bacterial membrane. In addition, IL-26 stimulation of macrophages and dendritic cells induces the secretion of tumor necrosis factor-α and inflammatory chemokines, contributing to antimicrobial defense [78]. Consistently, IL-26 produced by CD68⁺ alveolar macrophages and Th17 cells has been shown to strengthen pulmonary antimicrobial immunity by promoting neutrophil recruitment to sites of bacterial infection [21].
Through classical cytokine signalling via the IL-20R1/IL-10R2 receptor complex on non-hematopoietic cells, IL-26 also promotes the production of chemokines and antimicrobial peptides that enhance epithelial barrier defenses [26, 21].
In contrast, IL-26 can exert pro-inflammatory effects through a receptor-independent mechanism linked to its ability to form complexes with DNA with different origin. Their uptake in cells activates intracellular nucleic acid sensors such as TLR9, STING, and inflammasome pathways, leading to robust type I interferon and pro-inflammatory cytokine production. This mechanism provides a plausible explanation for the pathogenic role of IL-26 observed in chronic inflammatory conditions such as Crohn’s disease and in settings of sterile inflammation [53]. These findings show that the biological outcome of IL-26 activity is highly cell-type dependent.
Furthermore, another explanation may lie in the temporal context of IL-26 expression. Transient IL-26 production during acute infection may facilitate microbial clearance, whereas chronic overexpression or sustained tissue damage, accompanied by continuous release of host-derived DNA, may perpetuate a DNA-carrier-driven inflammatory loop characteristic of autoimmune and chronic inflammatory diseases.
Sources are provided in the References section of the manuscript
Comment 12: The authors could also mention the role of in-silico methods in the studies on IL-26.
Response 12: In the newly added Section 3, In silico methods in studies of IL-26, we provide a concise summary of key in silico approaches to predict the structure and physicochemical properties of IL-26, as well as on the integration of genomics and machine-learning methodologies. Literature sources are provided in the manuscript
Key applications of in silico methods for characterizing IL-26 include predicting structure and physicochemical properties, reconstructing interaction pathways, integrating omics data, and identifying expression quantitative trait loci, .
A sequence alignment using the ClustalW software revealed a higher similarity between IL-26 and IL-10 (26%) compared to IL-19 (20%) and IL-22 (21%). Thus, a structural model of IL-26 has been developed by in silico modeling with the existing IL-10 crystal 3D structure used as a starting point. The proposed 3D layout of the IL-26 molecule was obtained using the Modeler software and subsequently verified by Procheck. Further in silico analyses were performed to determine physicochemical properties of the IL-26 molecule, like surface charge densities (Adaptive Poisson-Boltzmann Solver), putative DNA-binding sites (MetaDBsite), and amphipathic helices (AmphipaSeek). A comprehensive map of IL-26 signaling has been constructed involving seven activation or inhibition interactions, 16 catalytic steps, 33 gene expression control points, 25 protein expression types, two transport mechanisms, and three molecular associations. A genome-wide genotyping has identified cis-expression quantitative trait loci associated with IBD-risk genes and immunity. Among them, an inflammation-dependent polymorphism, rs12582553, is located in close proximity to the IL-26 gene.
A recent study has integrated genomics and radiomics data using machine learning algorithms. This led to the development of a predictive model for colorectal cancer metastasis, in which IL-26 expression at both the mRNA and protein levels serves as an essential marker.
Comment 13: The conclusion is clear but could also include some future perspectives, therapeutic perspectives, unanswered questions, hypothesis that need to be verified.
Response 13: As suggested, we discussed knowledge gaps and potential new directions of investigation. These points are presented in new fully rewritten Conclusion section.
IL-26 is a multifunctional cytokine whose role in immunity and disease is yet to be fully elucidated. Despite significant progress in defining the contribution of IL-26 across various pathological contexts, many aspects of its involvement in regulating immune responses to both external and internal stimuli remain unclear. For instance, the exact spatiotemporal patterns of IL-26 expression in both normal and pathological conditions are not fully understood. On the other hand, the ability of IL-26 receptor subunits to associate with other cytokines creates new opportunities to investigate the detailed structural mechanisms of these interactions. Moreover, one of the cytokines with which the subunits of the IL-26 receptor associate is another cytokine of the IL-10 family, IL-20 [148]. IL-20 elicits responses in its target cells through two types of membrane receptors, IL-20R1/IL-20R2 and IL-22R1/IL-20R2. Whether such alternative receptors exist for IL-26 as well is yet to be determined. The context-dependent duality of IL-26 constitutes a key unresolved question, as the mechanisms controlling the balance between its beneficial and deleterious effects, such as the tissue microenvironment, receptor expression, and immune cell composition, remain incompletely understood.
Further research efforts will determine whether IL-26 may serve as a biomarker. IL-26 levels have been shown to correlate with disease activity in several inflammatory conditions, suggesting its potential utility as a biomarker of Th17-driven inflammation and possibly as a predictor of response to biologic therapies. Experimental and methodological limitations include the absence of IL-26 in rodents, which limits the use of conventional mouse models. The expanded application of single-cell and multi-omics approaches to identify IL-26-responsive cell populations could overcome these limitations and will be critical for advancing both mechanistic understanding and the translational potential of IL-26 biology. Crucial aspects that require further validation include whether DNA-binding is an essential physiological function of IL-26 or part of a pathological amplification loop.
Addressing these questions, together with antimicrobial and antiviral properties of IL-26 and its role in the development of diverse inflammatory and autoimmune diseases, will further establish IL-26 as a cytokine of considerable scientific interest and a potential therapeutic target for a range of disorders.
Reviewer 2 Report
Comments and Suggestions for Authors
This manuscript reported many aspects of IL-26, including gene, receptor-independent mechanism, cellular source, and multifaceted roles. This comprehensive review was suggested to be accepted after minor revisions.
- There were many abbreviations in the manuscript. Therefore, please provide these abbreviations the appropriate section of this manuscript, which was shown in the template.
- Please give the abbreviation when the full name appeared for the first time in the manuscript. For instance, ‘rheumatoid arthritis’ appeared in the Abstract(P1L22); however, its abbreviation ‘RA’ appeared in the subtitle ‘11. IL-26 in Rheumatoid arthritis (RA) and other inflammatory arthritides’(P13L474).
- There were some grammar or typo errors. For instance, ‘Multifaced role’ → ‘Multifaced roles’.
Author Response
Comment 1: There were many abbreviations in the manuscript. Therefore, please provide these abbreviations the appropriate section of this manuscript, which was shown in the template.
Response: We thank the reviewer for this suggestion. In accordance with the journal template, we have now added a dedicated Abbreviations section to the manuscript, in which all abbreviations used throughout the text are listed and defined.
Comment 2: Please give the abbreviation when the full name appeared for the first time in the manuscript. For instance, ‘rheumatoid arthritis’ appeared in the Abstract(P1L22); however, its abbreviation ‘RA’ appeared in the subtitle ‘11. IL-26 in Rheumatoid arthritis (RA) and other inflammatory arthritides’(P13L474).
Response: We have now revised the manuscript to ensure that all abbreviations are defined at their first occurrence in the text. Specifically, the abbreviation rheumatoid arthritis (RA) is now provided when the term first appears, and this correction has been applied consistently throughout the manuscript.
Comment 3: There were some grammar or typo errors. For instance, ‘Multifaced role’ → ‘Multifaced roles’.
Response: The manuscript has now been carefully revised for grammar and typographical errors, and language corrections were performed using the editing AI tool Grammarly. The title of the manuscript has been corrected as follows: “Multifaceted roles of IL-26 in physiological and pathological conditions”). We also corrected other minor grammar and spelling issues throughout the text.
Reviewer 3 Report
Comments and Suggestions for Authors
This review systematically summarizes the biological characteristics, immunomodulatory functions, and roles of IL-26 in various diseases. It is well-structured and covers a wide range of literature, providing an important reference for a deeper understanding of the pathophysiological mechanisms mediated by IL-26. However, there is still room for improvement in content organization, mechanism explanation, and information completeness. Specific revision suggestions are as follows:
1. It is recommended to add a table or diagram in the "6. Cellular source of IL-26" section to systematically summarize the main cell types that produce IL-26 in different tissues, enhancing readers' systematic understanding of the expression distribution characteristics of IL-26.
2. The "4. Receptor-independent mechanisms" section currently only lists potential action pathways but has not clarified the functional correlations and relative contributions among various mechanisms (such as the ability to bind to glycosaminoglycans and membrane penetration characteristics). It is suggested to integrate existing research data, conduct comparative analysis, and clarify the possible priority of action and potential for synergy.
3. In "15. IL-26 in cancer development", the phenomenon that IL-26 exhibits both pro-cancer and anti-cancer functions in different types of cancer is still lacking in mechanism-level analysis. It is suggested to discuss this in combination with differences in the tumor microenvironment (such as inflammatory background and immune infiltration status) and the activation patterns of key signaling pathways, revealing the potential basis for functional heterogeneity.
4. Figures 2 (Cellular sources of IL-26) and 3 (Target cells of IL-26) do not indicate the literature support for each cell type. It is recommended to add the key reference numbers in the figure captions to enhance the traceability and scientific credibility of the data presentation.
5. The review's conclusion does not systematically summarize the core limitations in the current IL-26 research field, such as the unclear alternative receptor and the lack of systematic depiction of spatiotemporal expression dynamics. It is suggested to add a "Research Prospects" section to systematically sort out knowledge gaps and propose future research directions.
6. Some disease-related sections (such as the association between IL-26 and COVID-19) can be supplemented with the latest research results published after 2023, especially those based on large-scale clinical cohorts or mechanism verification studies, to enhance the timeliness and academic frontier nature of the review.
7. There is controversy over whether monocytes express IL-26, with some studies reporting extremely low expression levels, while others observe constitutive expression. It is suggested to conduct an in-depth analysis of such discrepancies, exploring possible influencing factors such as differences in detection methods (mRNA vs protein level), sample processing conditions, or cell activation states, thereby enhancing the accuracy and interpretability of the information.
8. Some parts of the text have a high repetition rate, especially in the description of the basic characteristics of IL-26 in multiple chapters. It is recommended to integrate the language and adjust the structure of the entire text to avoid repetitive narration and enhance the compactness and logical coherence of the writing.
Author Response
Comment 1: It is recommended to add a table or diagram in the "6. Cellular source of IL-26" section to systematically summarize the main cell types that produce IL-26 in different tissues, enhancing readers' systematic understanding of the expression distribution characteristics of IL-26.
Response 1: A summary table has been added to Section 6 in the revised manuscript to systematically present the major IL-26–producing cell types across different tissues.
Table 2. IL-26-producing cell populations across various tissues.
|
IL-26 cell source |
Tissue |
Condition |
References |
|
Monocytes |
PBMCs |
Non-infected by Mycobacterium tuberculosis monocytes |
Guerra-Laso et al., 2015 |
|
Neoutrophiles |
Lesional skin biopsies |
Psoriasis |
Baldo et al., 2024 |
|
Macrophages |
Blood samples; IBD mucosa |
Rheumatoid arthritis; IBD |
Corvaisier et al., 2012; Fujii et al, 2017 |
|
Th17 |
Inflamed colonic lesions; Lesional human psoriatic skin |
Crohn’s disease; Psoriasis |
Dambacher et al., 2009; Wilson et al., 2007 |
|
NK |
IBD mucosa; mucosa-associated lymphoid tissues; PBMCs |
IBD; secondary lymphoid tissue |
Fujii et al. 2017; Cella et al., 2009; Wolk et al., 2002 |
|
NK T cells |
mononuclear cell isolates |
tuberculous pleurisy |
Zhang et al., 2019 |
|
CD3+, CD4+, CD8+ |
Serum; skin section |
bullous pemphigoid; ulcerative colitis |
Mizuno et al., 2022; Corridoni et al., 2020; Larochette et al., 2019 Fujii et al, 2017 (CD4+); Che et al., 2014 (CD8+) |
|
B cells |
Cell lines |
Human herpesvirus infected-B cells |
Knappe et al., 2000; Hummelshoj et al., 2006 |
|
Primary Bronchial Epithelial Cells |
bronchial brush biopsy |
Stimulation with TLR3 agonist poly-IC |
Che et al., 2017 |
|
Fibroblast-like synoviocytes |
Fibroblast-like synoviocytes cell line |
Rheumatoid arthritis |
Corvaisier et al., 2012 |
|
Smooth muscle cells |
Human pulmonary artery smooth muscle cells; smooth muscle cells |
Anti-neutrophil cytoplasmic Ab-associated vasculitis (AAV); spondyloarthritis |
Poli et al., 2017; Heftdal et al., 2017 |
|
Myofibroblast |
Colonic myofibroblasts; Smooth muscle actin expressing myofibroblasts |
IBD; spondyloarthritis |
Fujii et al, 2017; Heftdal et al, 2017 |
Comment 2: The "4. Receptor-independent mechanisms" section currently only lists potential action pathways but has not clarified the functional correlations and relative contributions among various mechanisms (such as the ability to bind to glycosaminoglycans and membrane penetration characteristics). It is suggested to integrate existing research data, conduct comparative analysis, and clarify the possible priority of action and potential for synergy.
Response 2: In the revised manuscript, Section 4 “Receptor-independent mechanisms” has been merged with Section 5 “IL-26 as a DNA-binding cytokine” under the original title of Section 4. We added mechanistic insights and functional links (lines 165-242) to compare the receptor-independent IL-26 signaling and clarify the potential of synergy via STING.
Comment 3: In "15. IL-26 in cancer development", the phenomenon that IL-26 exhibits both pro-cancer and anti-cancer functions in different types of cancer is still lacking in mechanism-level analysis. It is suggested to discuss this in combination with differences in the tumor microenvironment (such as inflammatory background and immune infiltration status) and the activation patterns of key signaling pathways, revealing the potential basis for functional heterogeneity.
Response 3: We thank the respected reviewer for pointing out this significant shortcoming of the manuscript. We provided the following explanatory text to clarify the potentially controversial association between IL-26 and cancer (section "15. IL-26 in cancer development", lines 781-833) .
The contradictory role of IL-26 in cancer may be partially explained by its essential role in autophagy. Increased levels of IL-26 in hepatocellular carcinoma (HCC) have been correlated with both unfavorable and favorable prognoses [105, Zhao]. In general, autophagy is believed to suppress cancer development at an early stage. However, during cancer progression, the elevated autophagic activity often promotes cancer survival and proliferation [Zhao]. Consistent with this assumption, both stimulation and suppression of autophagy have been established as therapeutic approaches for emerging and advanced malignancies, respectively [Huang, Nagelkerke]. Furthermore, IL-26 protein levels in HCC strongly depend on the disease stage [Zhao]. More importantly, in HCC, IL-26 has been mechanistically implicated in autophagy, presumably via a JAK/STAT3-dependent mechanism. In HCC cell lines, IL-26 treatment increases the number of autophagic vacuoles and LC3-II expression, both of which are crucial for autophagosome formation. In contrast, the opposite tendency has been observed for SQSTM1/p62 and Bcl2 following IL-26 treatment [Zhao]. The tumor microenvironment (TME) significantly modulates the impact of IL-26 on tumor formation and progression. Extensive evidence supports the overexpression of IL-26 in inflammatory contexts and the contribution of IL-26-induced cytokines, including IL-6, IL-8, and CXCL1, to TNBC metastasis [30]. In TME, exposure to IL-26 triggers an EGFR-TKI bypass via AKT- and JNK-mediated signaling, to promote TKI resistance. Moreover, the IL-26 influence on EphA3 and EGFR signaling suppresses ER stress pathways in TNBC [30]. The markedly higher IL-26 levels in HCC patients at least partially originate from macrophages residing in the tumor mass, as suggested by a significant overlap between IL-26 and CD68 immunostaining [Zhao]. In general, inflammation triggered by infiltrating innate immune cells contributes to tumor growth and metastasis [61, Shalapour]. Upon entering the TME and subsequent activation, infiltrating immune cells amplify inflammatory signaling, thereby directly contributing to tumor proliferation and metastasis [61, Shalapour, Korkaya, Park, Wculek, Su]. In this respect, TNBC tissues are often markedly infiltrated by immune cells, particularly T cells, neutrophils, and macrophages [61, Pistelli, Wei, Matsumoto]. Consistent with the previous findings, an elevated number of TNBC-infiltrating CD8+ T cells may indicate a favorable prognosis and chemotherapy sensitivity. In contrast, the presence of tumor-infiltrating macrophages has been associated with poor disease outcome [Matsumoto].
IL-26 suppresses the growth of intestinal epithelial cells [35, 43], although it promotes the proliferation of gastric tumor cells [35, Harris]. The primary cellular source of IL-26, Th17 cells, is evidently linked to gastric cancer [104, Yamada]. The presence of infiltrating Th17 cells in malignant gastric tissues may result from differentiation of CD4+ T cells or from migration of Th17 cells from the bloodstream into tumor tissues under the influence of the TME [104, Nordlohne, Majchrzak]. Th17 cells are known for their complex roles in various malignancies, facilitating tumor progression through advancing angiogenesis and immune suppression. Meanwhile, Th17 may also promote antitumor immune responses by enhancing the activity of CD8+ T cells [Sutanto]. In tumors, Th17 cells exhibit remarkable plasticity, acquiring phenotypes such as Th1 cells or regulatory T cells in response to the TME. Their plasticity is essential for their function within the TME, which can modulate Th17 activity through cytokines, engagement of signaling cascades, and epigenetic mechanisms [Sutanto]. A recent study of patients with colorectal cancer found significantly higher IL-26 RNA and protein levels in the non-metastatic group than in the metastatic group. Additionally, the IL-26 high-expression group demonstrated improved overall and disease-free survival. Moreover, the pool of differentially expressed genes positively correlating with high IL-26 levels was enriched for JAK/STAT and epsilon receptor signaling [108]. Interleukin-mediated JAK/STAT activation, subsequently causing anti-apoptotic but not anti-inflammatory or proliferative effects in primary colonic epithelial cells, has been previously described for another interleukin, IL-11 [Kiesling]. Taken together, the existence of such non-metastatic and non-proliferative associations of IL-26 and JAK/STAT signaling further complicates their potential role in cancer progression, cellular proliferation, and apoptosis in the context of TME.
Comment 4: Figures 2 (Cellular sources of IL-26) and 3 (Target cells of IL-26) do not indicate the literature support for each cell type. It is recommended to add the key reference numbers in the figure captions to enhance the traceability and scientific credibility of the data presentation.
Response 4:In the revised manuscript, key reference numbers supporting the target cells of IL-26 have been added to the Figure 3 caption. The literature supporting the cellular sources of IL-26 is comprehensively summarized in Table 1, which was added in response to Comment 1. To avoid redundancy, we did not repeat the same reference information in the captions of Figure 2 but instead direct readers to the corresponding Table 2 where the supporting literature is systematically presented.
Figure 3. Target cells of IL-26. Target cells include neutrophils (Muzino et al., 2022), M1 macrophages (Lin et al., 2020), monocytes and memory CD4⁺ T cells (Corvaisier et al., 2012), dendritic cells (Hawerkamp et al., 2020), vascular endothelial cells (Hatano et al., 2019), keratinocytes and epithelial cells (Hor et al., 2004), chondrocytes (Chen et al., 2021), fibroblast-like synoviocytes and osteoclasts (Lee et al., 2019), as well as subepithelial myofibroblasts (Fujii et al., 2017).
Comment 5: The review's conclusion does not systematically summarize the core limitations in the current IL-26 research field, such as the unclear alternative receptor and the lack of systematic depiction of spatiotemporal expression dynamics. It is suggested to add a "Research Prospects" section to systematically sort out knowledge gaps and propose future research directions.
Response 5: As suggested by the reviewer, we added an entire section entitled “Research gaps and future perspectives”. In it, we discuss knowledge gaps and potential new directions of investigation.
IL-26 is a multifunctional cytokine whose role in immunity and disease is yet to be fully elucidated. Despite significant progress in defining the contribution of IL-26 across various pathological contexts, many aspects of its involvement in regulating immune responses to both external and internal stimuli remain unclear. For instance, the exact spatiotemporal patterns of IL-26 expression in both normal and pathological conditions are not fully understood. On the other hand, the ability of IL-26 receptor subunits to associate with other cytokines creates new opportunities to investigate the detailed structural mechanisms of these interactions. Moreover, one of the cytokines with which the subunits of the IL-26 receptor associate is another cytokine of the IL-10 family, IL-20. IL-20 elicits responses in its target cells through two types of membrane receptors, IL-20R1/IL-20R2 and IL-22R1/IL-20R2 [1]. Whether such alternative receptors exist for IL-26 as well is yet to be determined. The context-dependent duality of IL-26 constitutes a key unresolved question, as the mechanisms controlling the balance between its beneficial and deleterious effects, such as the tissue microenvironment, receptor expression, and immune cell composition, remain incompletely understood.
Further research efforts will determine whether IL-26 may serve as a biomarker. IL-26 levels have been shown to correlate with disease activity in several inflammatory conditions, suggesting its potential utility as a biomarker of Th17-driven inflammation and possibly as a predictor of response to biologic therapies. Experimental and methodological limitations include the absence of IL-26 in rodents, which limits the use of conventional mouse models. The expanded application of single-cell and multi-omics approaches to identify IL-26-responsive cell populations could overcome these limitations and will be critical for advancing both mechanistic understanding and the translational potential of IL-26 biology. Crucial aspects that require further validation include whether DNA-binding is an essential physiological function of IL-26 or part of a pathological amplification loop.
Addressing these questions, together with antimicrobial and antiviral properties of IL-26 and its role in the development of diverse inflammatory and autoimmune diseases, will further establish IL-26 as a cytokine of considerable scientific interest and a potential therapeutic target for a range of disorders.
Source: Logsdon, N.J.; Deshpande, A.; Harris, B.D.; Rajashankar, K.R.; Walter, M.R. Structural basis for receptor sharing and activation by interleukin-20 receptor-2 binding cytokines. Proc. Natl. Acad. Sci. USA 2012, 109, 12704–12709.
Comment 6: Some disease-related sections (such as the association between IL-26 and COVID-19) can be supplemented with the latest research results published after 2023, especially those based on large-scale clinical cohorts or mechanism verification studies, to enhance the timeliness and academic frontier nature of the review.
Response 6: According to the reviewer’s recommendation, we added two short paragraphs addressing the association between asthma and COVID-19 infection with an accent on the potential involvement of IL-26 in COVID-19. This newly added text presents recent findings and focuses on potential mechanistic pathways by which COVID-19 infection may predispose to allergic disease, with particular emphasis on the involvement of IL-26 in this process.
Epidemiological data from large patient cohorts have associated COVID-19 infection in children with the development of allergic predisposition [1]. Significant downregulation of the ACE2, which serves as a coronavirus receptor, has been observed in nasal and airway epithelial cells of patients with type 2 asthma and allergic rhinitis. Furthermore, the expression levels of ACE2 have been negatively associated with type 2 cytokines. As a result, ACE2 downregulation may impede viral infiltration in host cells, but at the same time, it may increase type 2 inflammatory response [2].
According to the current mechanistic model, the viral-induced epithelial damage causes a surge in alarmin cytokines (IL-25, IL-33, and TSLP), which stimulate the differentiation of naive CD4+ T cells into Th2 helpers and the expansion of group 2 innate lymphoid cells. It also promotes long-lasting epigenetic changes and reprogramming of hematopoietic stem cells towards the Th2 phenotype. IL-7 and IL-15 alter T- and B-cell homeostasis through their roles in immune memory, while impaired T regulatory cell function may diminish immune tolerance [1]. In allergic airways , IL-33 often modulates Th2-mediated repair, promoting TGF-β and collagen accumulation, whereas in the non-allergic respiratory system, IL-33 may exacerbate the IFN-γ- driven cytokine storm [3]. Given the role of IFN-γ in that cytokine storm, it is noteworthy to mention that the IL-26 gene is located in close proximity to the IFN-γ gene and even shares an enhancer sequence with it [4]. Provided that IL-26 expression significantly correlates with the severity of COVID-19 infection [5,6], the potential functional involvement of IL-26 in COVID-19 can be suspected.
Sources:
[1] Filippatos, F.; Matara, D.-I.; Michos, A.; Kakleas, K. Immunological Mechanisms Underlying Allergy Predisposition After SARS-CoV-2 Infection in Children. Cells 2025, 14, 1511. https://doi.org/10.3390/cells14191511
[2] Kimura, H.; Francisco, D.; Conway, M.; Martinez, F. D.; Vercelli, D.; Polverino, F.; Billheimer, D.; Kraft, M. Type 2 inflammation modulates ACE2 and TMPRSS2 in airway epithelial cells. J Allergy Clin Immunol. 2020, 146, 80-88.e8. https://doi.org/10.1016/j.jaci.2020.05.004
[3] Murdoch, J.R.; Lloyd, C.M. Chronic inflammation and asthma. Mutat. Res. 2010, 690, 24–39.
[4] Larochette, V.; Miot, C.; Poli, C.; Beaumont, E.; Roingeard, P.; Fickenscher, H.; Jeannin, P.; Delneste, Y. IL-26, a Cytokine with Roles in Extracellular DNA-Induced Inflammation and Microbial Defense. Front. Immunol. 2019, 10, 204. https://doi.org/10.3389/fimmu.2019.00204
[5] Cardenas, E.I.; Robertson, J.; Misaghian, S.; Brown, J.; Wang, M.; Stengelin, M.; Sigal, G.; Wohlstadter, J.; Gisslén, M.; Lindén, A. Systemic Increase in IL-26 Is Associated with Severe COVID-19 and Comorbid Obstructive Lung Disease. Front. Immunol. 2024, 15, 1434186. https://doi.org/10.3389/fimmu.2024.1434186
[6] Cardenas, E.I.; Ekstedt, S.; Piersiala, K.; Petro, M.; Karlsson, A.; Kågedal, Å.; Kumlien Georén, S.; Cardell, L.O.; Lindén, A. Increased IL-26 Associates with Markers of Hyperinflammation and Tissue Damage in Patients with Acute COVID-19. Front. Immunol. 2022, 13, 1016991. https://doi.org/10.3389/fimmu.2022.1016991
Comment 7: There is controversy over whether monocytes express IL-26, with some studies reporting extremely low expression levels, while others observe constitutive expression. It is suggested to conduct an in-depth analysis of such discrepancies, exploring possible influencing factors such as differences in detection methods (mRNA vs protein level), sample processing conditions, or cell activation states, thereby enhancing the accuracy and interpretability of the information.
Response 7: We fully accepted this recommendation and provided an explanatory text, which discusses the potentially conflicting data regarding the IL-26 expression in monocytes. This newly added text addresses the controversies by listing discrepancies in sampling, cell stimulation protocols, and detection methods. The following text was included in the manuscript (lines 295-327):
Several studies have reported conflicting results regarding IL-26 expression in monocytes. For instance, two studies by Wolk et al. have suggested IL-26 expression is absent in monocytes (42, 54). However, according to other sources, IL-26 expression in monocytes is constitutive, although it is lower than in T cells (55). When considering activated monocytes specifically, IL-26 exhibits reduced expression in monocytes infected with Mycobacterium tuberculosis at the RNA level (55). Conversely, in monocytes exposed to lipopolysaccharides (LPS) and IFN-γ, combined with an anti-IL-10 antibody, leads to an increased production of IL-26 (56). Furthermore, alveolar mature macrophages isolated from the lungs of healthy individuals secrete IL-26 as a response to endotoxin exposure (25). Existing discrepancies across various studies may be potentially explained by different protocols for sample processing, dissimilar strategies for inducing cell activation, and, especially, different detection methods. For instance, in both studies by Wolk et al. (42, 54) and in the work of Guerra-Lasso et al., the authors apply magnetic cell separation (MACS) to purify CD14+ monocytes from white blood cell populations, while Nagalakshmi et al. utilize elutriation for the purpose of monocyte isolation. Meanwhile, Bolin et al. preferred immunocytochemistry (ICC) and immunocytofluorescence (ICF) combined with fluorescence-activated cell sorting (FACS) to isolate and characterize mature CD68+ macrophages from bronchoalveolar lavage samples. Discrepancies in the stimulation/differentiation procedures are also significant confounding factors: LPS from Escherichia coli, culturing in the presence of M-CSF, infection with Mycobacterium tuberculosis, LPS and IFN-γ paired with an anti-IL-10 antibody, and endotoxin stimulation. However, the factor of primary importance potentially affecting the rigor of experiments is the method of detection. For example, several studies rely only on RT-qPCR as a technique for quantitative evaluation of IL-26 levels. However, applying more than one quantitation methods, which produce comparable results, strengthens the reliability of the data obtained. Accordingly, IL-26 has been characterized in blood monocytes in the context of Mycobacterium tuberculosis infection by microarray, RT-qPCR, and ELISA, confirming the constitutive expression of IL-26 and its downregulation as a response to Mycobacterium infection. Meanwhile, the study by Bolin et al. provides solid evidence for the expression of IL-26 in mature alveolar macrophages based on the marked colocalization of IL-26 and CD68 after ICC, ICF, and FACS. These results receive further confirmation by ELISA and RT-qPCR.
Comment 8: Some parts of the text have a high repetition rate, especially in the description of the basic characteristics of IL-26 in multiple chapters. It is recommended to integrate the language and adjust the structure of the entire text to avoid repetitive narration and enhance the compactness and logical coherence of the writing.
Response 8: The entire manuscript was thoroughly revised to integrate information and avoid unnecessary repetition.
Round 2
Reviewer 1 Report
Comments and Suggestions for Authors
The authors have revised and improved their work. This version can be accepted.
Reviewer 3 Report
Comments and Suggestions for Authors
No additional comments.